# Research on High Precision Stiffness Modeling Method of Redundant Over-Constrained Parallel Mechanism

**DOI:** 10.3390/s23135916

**Published:** 2023-06-26

**Authors:** Sen Wang, Haoran Li, Xueyan Han, Jiahao Wei, Tao Zhang, Shihua Li

**Affiliations:** 1Parallel Robot and Mechatronic System Laboratory of Hebei Province, Yanshan University, Qinhuangdao 066004, China; 2College of Mechanical Engineering, Yanshan University, Qinhuangdao 066004, China

**Keywords:** redundant parallel mechanism, stiffness, joint clearance, joint contact deformation, probability statistical model

## Abstract

Traditional stiffness modeling methods do not consider all factors comprehensively, and the modeling methods are not unified, lacking a global stiffness model. Based on screw theory, strain energy and the virtual work principle, a static stiffness modeling method for redundant over-constrained parallel mechanisms (PMs) with clearance was proposed that considers the driving stiffness, branch deformation, redundant driving, joint clearance and joint contact deformation. First, the driving stiffness and branch deformation were considered. According to the strain energy and Castiliano’s second theorem, the global stiffness matrix of the ideal joint mechanism was obtained. The offset of the branch was analyzed according to the restraint force of each branch. The mathematical relationship between the joint clearance and joint contact deformation and the end deformation was established. Based on the probability statistical model, the uncertainty of the offset value of the clearance joint and the contact area of the joint caused by the coupling of the branch constraint force was solved. Finally, taking a 2UPR-RR-2RPU redundant PM as an example, a stiffness simulation of the mechanism was carried out using the finite element method. The research results show that the high-precision stiffness modeling method proposed in this paper is correct, and provides an effective method for evaluating the stiffness performance of the PM.

## 1. Introduction

The stiffness reflects the displacement of the end of the mechanism relative to the expected position under external forces. The stiffness of the parallel mechanism (PM) often determines the kinematic positioning accuracy and kinematic characteristics of the end effector. In addition, insufficient stiffness may reduce the natural frequency and dynamic performance of the system. Therefore, the stiffness performance should be evaluated in the product design phase.

In terms of stiffness modeling methods of PM, the main methods include screw theory [1], the semi-analytical method [2], the virtual joint method [3], strain energy [4,5], the structural stiffness matrix method [6], etc. The factors considered in the modeling method are gradually developing toward moving platform flexibility [7], driving stiffness and branch flexibility [8,9] and gravity [10,11,12], making the accuracy of analytical model building more accurate and closer to perfection. Compared with non-redundant mechanisms, redundant mechanisms have many advantages, such as avoiding kinematic singularity, improving stiffness distribution, enhancing bearing capacity and improving dynamic characteristics. Therefore, redundant driving branches are usually added to the mechanism to improve the stiffness distribution in its workspace [13,14,15,16,17].

When the mechanism has clearance joints, the end position depends on the attitude, load, geometric parameters of the joint and the joint force, and the relative position inside the joint is uncertain. In order to solve this problem, the commonly used methods are the worst-case method, probability statistical model, interval analysis method, virtual connection method and hybrid dimension reduction method. Choi et al. [18] put forward a probability model for the output error analysis of a four-bar mechanism considering the joint clearance and its tolerance effect. Frisoli [19] maximized the attitude error function by combining the analytical solution with the numerical solution. The worst-case angular position accuracy and linear position accuracy of the parallel mechanism with clearance were determined. Wang et al. [20] proposed a hybrid dimension reduction method for the motion reliability analysis of a slider crank mechanism with clearance. Yao [21] analyzed the error space of a Stewart mechanism with an interval analysis method. Chouaibi [22] established the mathematical model between the end error of RAF PM and the external load, structural parameters, attitude and joint clearance through the principle of virtual work. Zhang [23] regarded the joint clearance as a massless virtual connection and obtained the error boundary and distribution of the 3-RPR PM. Cammarata [24] proposed a new method for solving the displacement and rotation of clearance joint nodes of over-constrained mechanisms. It was used to determine the position and direction errors of over-constrained mechanisms with clearance. Ding [25] modeled and numerically estimated the attitude error of 3-RPR PM caused by clearance and branch deformation from the perspective of inverse kinematics.

The static stiffness model that considers the joint clearance needs to study the linear deformation and angular deformation of the end in all directions. In particular, the deformation of the end constraint direction needs to be considered. 

In order to establish a static stiffness model of the mechanism considering hinge clearance, it is necessary to study the line deformation and angular deformation in each direction of the end, considering the deformation in the direction of the end constraint in particular. Most of the existing error models are based on inverse kinematics, which can only obtain the deformation in the direction of the end output DOF, and they are not suitable for stiffness models. The over-constrained PM is affected by the coupling of the constraint force and constraint couple at the joint. The line deflection and angular deflection of the joint with clearance in the branch is coupled. The offset/deflection of the joint with clearance is affected by multiple factors; for example, the clearance size, external load, mechanism attitude, redundant branch and the constraint force/torque amplitude of the branch. This makes the modeling and analysis of its clearance model more difficult. For PMs used in a heavy-load field, the constraint force between joints is large. According to Hertz contact theory, the greater the contact force between two objects in contact, the greater the contact deformation of the hinge.

To summarize, previous stiffness studies focused on local factors, resulting in inaccurate stiffness modeling. This paper studied the position distribution of the clearance joints under the branch-coupling constraint of redundant over-constrained mechanisms under different external loads. The mathematical relationships between the driving stiffness, branch deformation, redundant driving, joint clearance, joint contact deformation and end stiffness of the mechanism were established. The influence of each factor on the stiffness of the mechanism was analyzed. A novel 2UPR-RR-2RPU redundant over-constrained PM was taken as an example to verify the correctness of the proposed model.

## 2. High-Precision Stiffness Modeling Method of Redundant Over-Constrained PMs Considering Multiple Factors

### 2.1. High-Precision Stiffness Modeling Process of Redundant Over-Constrained PMs Considering Multiple Factors

High-precision stiffness modeling method of redundant over-constrained PMs considering multiple factors:(1)According to the screw theory, the drive screw and constraint screw of a redundant branch and non-redundant branch are analyzed. Then, the drive Jacobian matrix and constraint Jacobian matrix are constructed.(2)The driving stiffness is analyzed from the perspective of material mechanics. The strain energy of each branch under the constraint force is calculated and the functional relationship between the strain energy of the branch and the constraint force of the branch is derived.(3)Based on Castigliano’s second theorem, the partial derivatives of the strain energy of each branch on the constraint force of the branch are calculated, respectively. The functional relationship between the deformation of the end of the branch and the constraint force of the branch is obtained, namely the flexibility matrix of the branch.(4)According to the deformation compatibility equation of the mechanism and virtual work principle, the stiffness matrix of the whole mechanism and the constraint force of each branch are solved and the end deformation caused by the driving stiffness and branch deformation is obtained.(5)Based on the constraints of each branch, the position distribution of the clearance joints and the contact area of the joints are predicted by the probability statistical model. It solves the problem of the modeling error caused by coupling constraints of over-constrained mechanisms. The deformation caused by the joint clearance and joint contact deformation along the constraint direction at the connection between each branch and the moving platform end is analyzed. Based on the principle of virtual work, the end deformation interval caused by joint clearance and joint contact deformation is obtained.(6)Assuming that there is no coupling between rigid body displacement and elastic deformation, the four factors are regarded as mutually independent and can be linearly superposed. A high-precision static stiffness model considering multiple factors is obtained.

### 2.2. General Equation of High-Precision Stiffness Model of Redundant Over-Constrained PMs Considering Multiple Factors

Based on the above assumptions and modeling process, the high-precision static stiffness model considering multiple factors can be expressed as
(1)$$\begin{array}{l} \Delta X=\Delta X^{\varsigma }+\Delta X^{\mathrm{cle}}+\Delta X^{\mathrm{con}}\\ ={\left(k_{a,c}\right)}^{-1}w+J_{\mathrm{nac}}^{*}K_{\mathrm{nac}}^{\mathrm{cle}}+J_{\mathrm{rac}}^{*}K_{\mathrm{rac}}^{\mathrm{cle}}+J_{\mathrm{nac}}^{*}K_{\mathrm{nac}}^{\mathrm{con}}\tau_{\mathrm{nac}}+J_{\mathrm{rac}}^{*}K_{\mathrm{rac}}^{\mathrm{con}}\tau_{\mathrm{rac}} \end{array}$$

In the Equation, ΔX represents the total deformation interval in the direction of six DOFs of the end, and $\Delta X^{\varsigma }$ represents the end deformation caused by the driving stiffness, branch deformation and redundant driving. $\Delta X^{\mathrm{cle}}$ represents the end deformation interval caused by the joint clearance corrected based on the statistical model. $\Delta X^{\mathrm{con}}$ represents the end deformation interval caused by the joint contact deformation corrected based on the statistical model. w represents the six-dimensional load of the moving platform.
(2)$$\begin{array}{l} J_{\mathrm{nac}}^{*}=\mathrm{diag}\left(\begin{array}{cc} {\left(J_{\mathrm{na}}^{*}\right)}^{T} & {\left(J_{\mathrm{nc}}^{*}\right)}^{T} \end{array}\right),J_{\mathrm{rac}}^{*}=\mathrm{diag}\left(\begin{array}{cc} {\left(J_{\mathrm{ra}}^{*}\right)}^{T} & {\left(J_{\mathrm{rc}}^{*}\right)}^{T} \end{array}\right)\\ K_{\mathrm{nac}}^{\mathrm{cle}}=\left[\begin{array}{c} K_{\mathrm{na}}^{\mathrm{cle}}\\ K_{\mathrm{nc}}^{\mathrm{cle}} \end{array}\right],K_{\mathrm{rac}}^{\mathrm{cle}}=\left(\begin{array}{c} K_{\mathrm{ra}}^{\mathrm{cle}}\\ K_{\mathrm{rc}}^{\mathrm{cle}} \end{array}\right)\\ K_{\mathrm{nac}}^{\mathrm{con}}=\mathrm{diag}\left(\begin{array}{cc} K_{\mathrm{na}}^{\mathrm{con}} & K_{\mathrm{nc}}^{\mathrm{con}} \end{array}\right),K_{\mathrm{rac}}^{\mathrm{con}}\mathrm{diag}\left(\begin{array}{cc} K_{\mathrm{ra}}^{\mathrm{con}} & K_{\mathrm{rc}}^{\mathrm{con}} \end{array}\right)\\ \tau_{\mathrm{nac}}=\left[\begin{array}{c} \left|\tau_{\mathrm{na}}\right|\\ \left|\tau_{\mathrm{nc}}\right| \end{array}\right],\tau_{\mathrm{rac}}=\left[\begin{array}{c} \left|\tau_{\mathrm{ra}}\right|\\ \left|\tau_{\mathrm{rc}}\right| \end{array}\right] \end{array}$$

In the Equation, $J_{\mathrm{na}}^{*}$ represents the mapping matrix of the non-redundant branch drive force and external load. $J_{\mathrm{nc}}^{*}$ represents the mapping matrix of the non-redundant branch constraint force and external load. $J_{\mathrm{ra}}^{*}$ represents the mapping matrix of the redundant branch drive force and external load. $J_{\mathrm{rc}}^{*}$ represents the mapping matrix of the redundant branch constraint force and external load. $K_{\mathrm{na}}^{\mathrm{cle}}$ represents the offset matrix of the end of the branch caused by the driving force of the non-redundant branch with clearance. $K_{\mathrm{nc}}^{\mathrm{cle}}$ represents the offset matrix of the end of the branch caused by the constraint force of the non-redundant branch with clearance. $K_{\mathrm{ra}}^{\mathrm{cle}}$ represents the offset matrix of the end of the branch caused by the driving force of the redundant branch with clearance. $K_{\mathrm{rc}}^{\mathrm{cle}}$ represents the offset matrix of the end of the branch caused by the constraint force of the redundant branch with clearance. $K_{\mathrm{na}}^{\mathrm{con}}$ represents the contact stiffness matrix of the joint at the end of the branch caused by the driving force of the non-redundant branch. $K_{\mathrm{nc}}^{\mathrm{con}}$ represents the contact stiffness matrix of the joint at the end of the branch caused by the restraint force of the non-redundant branch chain. $K_{\mathrm{ra}}^{\mathrm{con}}$ represents the contact stiffness matrix of the joint at the end of the branch caused by the driving force of the redundant branch. $K_{\mathrm{rc}}^{\mathrm{con}}$ represents the contact stiffness matrix of the joint at the end of the branch caused by the restraint force of the redundant branch chain. $\tau_{\mathrm{na}}$ represents the non-redundant branch driving force matrix. $\tau_{\mathrm{nc}}$ represents the non-redundant branch constraint matrix. $\tau_{\mathrm{ra}}$ represents the redundant branch driving force matrix. $\tau_{\mathrm{rc}}$ represents the redundant branch constraint matrix.

### 2.3. Stiffness Matrix Derivation and Restraint Force Analysis of Redundant PMs with Ideal Joints

The force analysis of over-constrained PMs is statically indeterminate, and the structural stiffness of each branch needs to be considered. In this paper, the moving platform was assumed to be rigid, and the spatial composite elastic deformation of the branch bending, stretching and torsion was considered. The driving force/restraint stiffness matrix of the branch is defined as the mapping relationship between the driving force/restraint amplitude and the elastic deformation of the end along the driving force/restraint direction. 

The moving platform of the mechanism is acted on by a six-dimensional external force ***w***. According to the static equilibrium equation of the moving platform, we can obtain
(3)$$w=J_{\mathrm{na}}^{T}\tau_{\mathrm{na}}+J_{\mathrm{ra}}^{T}\tau_{\mathrm{ra}}+J_{\mathrm{nc}}^{T}\tau_{\mathrm{nc}}+J_{\mathrm{rc}}^{T}\tau_{\mathrm{rc}}$$

$\tau_{\mathrm{na}}$, $\tau_{\mathrm{nc}}$ represent the column vectors composed of the driving force and constraint force amplitude of each non-redundant branch, respectively. $\tau_{\mathrm{na}}$, $\tau_{\mathrm{nc}}$ represent the column vectors composed of the driving force and constraint force amplitude of each redundant branch, respectively.
(4)$$J_{a}=\left[\begin{array}{c} J_{\mathrm{na}}\\ J_{\mathrm{ra}} \end{array}\right]={\left[{\$}_{a,1}^{T}\;\cdots \;{\$}_{a,k}^{T}\;\cdots \;{\$}_{a,i}^{T}\right]}^{T}$$(5)$$J_{c}=\left[\begin{array}{c} J_{\mathrm{nc}}\\ J_{\mathrm{rc}} \end{array}\right]={\left[{\$}_{c,1}^{T}\;\cdots \;{\$}_{c,n}^{T}\;\cdots \;{\$}_{c,m}^{T}\right]}^{T}$$

In Equations (4) and (5), ${\$}_{a,i}$ represents the driving force screw of branch *i* acting on the moving platform, and ${\$}_{c,i}$ represents the constraint force screw of branch *i* acting on the moving platform. *n* represents the number of all constrained screws of non-redundant branches in the mechanism, *m*, and *i* represents the number of all constraint forces of redundant branches in the mechanism.

Assuming that the branch driving force $\tau_{\mathrm{ai}}$ and the branch terminal deformation $\Delta q_{\mathrm{ai}}$ caused by the driving force and the branch restraint force $\tau_{\mathrm{ci}}$ and the deformation $\Delta q_{\mathrm{ci}}$ of the branch terminal caused by the constraint force satisfy the basic linear elasticity assumption, we can obtain
(6)$$\tau_{\mathrm{na}}=k_{\mathrm{na}}\Delta q_{\mathrm{na}};\tau_{\mathrm{ra}}=k_{\mathrm{ra}}\Delta q_{\mathrm{ra}}$$
(7)$$\tau_{\mathrm{nc}}=k_{\mathrm{nc}}\Delta q_{\mathrm{nc}};\tau_{\mathrm{rc}}=k_{\mathrm{rc}}\Delta q_{\mathrm{rc}}$$
where $k_{\mathrm{na}}=\mathrm{diag}\left(\begin{array}{cccc} k_{a1} & k_{a2} & \cdots & k_{ak} \end{array}\right)$, $k_{\mathrm{nc}}=\mathrm{diag}\left(\begin{array}{cccc} k_{c1} & k_{c2} & \cdots & k_{\mathrm{cn}} \end{array}\right)$ and $k_{\mathrm{aj}}(1\leq j\leq k)$ are defined as the stiffness of the non-redundant branch in the direction of the driving force. $k_{cj}(1\leq j\leq n)$ is defined as the stiffness of the non-redundant branch in the direction of the constraint force. $k_{\mathrm{ra}}=\mathrm{diag}\left(\begin{array}{cccc} k_{ak} & k_{a(k+1)} & \cdots & k_{\mathrm{ai}} \end{array}\right)$, $k_{\mathrm{rc}}=\mathrm{diag}\left(\begin{array}{cccc} k_{cn} & k_{c(n+1)} & \cdots & k_{cm} \end{array}\right)$ and $k_{\mathrm{aj}}(k\leq j\leq i)$ are defined as the stiffness of the redundant branch in the direction of the driving force. $k_{cj}(n\leq j\leq m)$ is defined as the stiffness of the redundant branch in the direction of the constraint force.
(8)$$w^{T}\Delta X^{\varsigma }=\tau_{\mathrm{na}}^{T}\Delta q_{\mathrm{na}}+\tau_{\mathrm{nc}}^{T}\Delta q_{\mathrm{nc}}+\tau_{\mathrm{ra}}^{T}\Delta q_{\mathrm{ra}}+\tau_{\mathrm{rc}}^{T}\Delta q_{\mathrm{rc}}$$

In Equation (8), $\Delta X^{\varsigma }={\left(\begin{array}{cccccc} \Delta x & \Delta y & \Delta z & \Delta \alpha & \Delta \beta & \Delta \gamma \end{array}\right)}^{T}$ represents the micro-deformation of the moving platform under the external forces when only the driving stiffness and branch deformation are considered.

Sorting Equation (3) and Equations (6)–(8) simultaneously, we can obtain
(9)$$\begin{array}{l} \tau_{\mathrm{na}}^{T}\left(J_{\mathrm{na}}\Delta X^{\varsigma }-\Delta q_{\mathrm{na}}\right)+\tau_{\mathrm{ra}}^{T}\left(J_{\mathrm{ra}}\Delta X^{\varsigma }-\Delta q_{\mathrm{ra}}\right)\\ +\tau_{\mathrm{nc}}^{T}\left(J_{\mathrm{nc}}\Delta X^{\varsigma }-\Delta q_{\mathrm{nc}}\right)+\tau_{\mathrm{rc}}^{T}\left(J_{\mathrm{rc}}\Delta X^{\varsigma }-\Delta q_{\mathrm{rc}}\right)=0 \end{array}$$

In order for Equation (9) to be established, it is necessary to satisfy
(10)$$J_{\mathrm{na}}\Delta X^{\varsigma }-\Delta q_{\mathrm{na}}=0;J_{\mathrm{ra}}\Delta X^{\varsigma }-\Delta q_{\mathrm{ra}}=0$$
(11)$$J_{\mathrm{nc}}\Delta X^{\varsigma }-\Delta q_{\mathrm{nc}}=0;J_{\mathrm{rc}}\Delta X^{\varsigma }-\Delta q_{\mathrm{rc}}=0$$

Combining Equations (3), (6), (7), (10) and (11), the relationship between the external force on the moving platform and the micro-deformation of the moving platform when considering the driving stiffness and the deformation of the branch can be obtained.
(12)$$\begin{array}{l} w=J_{\mathrm{na}}^{T}k_{\mathrm{na}}J_{\mathrm{na}}\Delta X^{\varsigma }+J_{\mathrm{ra}}^{T}k_{\mathrm{ra}}J_{\mathrm{ra}}\Delta X^{\varsigma }\\ +J_{\mathrm{nc}}^{T}k_{\mathrm{nc}}J_{\mathrm{nc}}\Delta X^{\varsigma }+J_{\mathrm{rc}}^{T}k_{\mathrm{rc}}J_{\mathrm{rc}}\Delta X^{\varsigma }\\ =k_{a,c}\Delta X^{\varsigma } \end{array}$$

In the Equation,
(13)$$k_{a,c}=\left(\begin{array}{l} \left[\begin{array}{cc} J_{\mathrm{na}}^{T} & J_{\mathrm{nc}}^{T} \end{array}\right]\left[\begin{array}{cc} k_{\mathrm{na}} & 0\\ 0 & k_{\mathrm{nc}} \end{array}\right]\left[\begin{array}{c} J_{\mathrm{na}}\\ J_{\mathrm{nc}} \end{array}\right]\\ +\left[\begin{array}{cc} J_{\mathrm{ra}}^{T} & J_{\mathrm{rc}}^{T} \end{array}\right]\left[\begin{array}{cc} k_{\mathrm{ra}} & 0\\ 0 & k_{\mathrm{rc}} \end{array}\right]\left[\begin{array}{c} J_{\mathrm{ra}}\\ J_{\mathrm{rc}} \end{array}\right] \end{array}\right)$$

$k_{a,c}$ represents the stiffness matrix of the over-constrained PM considering the driving stiffness of the mechanism and the deformation of the branches.

Combining Equations (5), (6) and Equations (9)–(11), the expressions of the driving force, constraint force and external force of the moving platform can be obtained.
(14)$$\tau_{\mathrm{ni}}=k_{\mathrm{ni}}J_{\mathrm{ni}}k_{a,c}^{-1}w;\;\tau_{\mathrm{ra}}=k_{\mathrm{ri}}J_{\mathrm{ri}}k_{a,c}^{-1}w$$
where subscript *a* represents the driving force, *c* represents the constraint force (*i* = *a*, *c*), *n* represents a basic branch and *r* represents a redundant branch. 

Rearranging the above equation can obtain
(15)$$\tau =\left[\begin{array}{c} \tau_{\mathrm{na}}\\ \tau_{\mathrm{ra}}\\ \tau_{\mathrm{nc}}\\ \tau_{\mathrm{rc}} \end{array}\right]=\left[\begin{array}{c} k_{\mathrm{na}}J_{\mathrm{na}}k_{a,c}^{-1}\\ k_{\mathrm{ra}}J_{\mathrm{ra}}k_{a,c}^{-1}\\ k_{\mathrm{nc}}J_{\mathrm{nc}}k_{a,c}^{-1}\\ k_{\mathrm{rc}}J_{\mathrm{rc}}k_{a,c}^{-1} \end{array}\right]w=\left[\begin{array}{c} J_{\mathrm{na}}^{*}\\ J_{\mathrm{ra}}^{*}\\ J_{\mathrm{nc}}^{*}\\ J_{\mathrm{rc}}^{*} \end{array}\right]w$$

### 2.4. Stiffness Modeling Method of Over-Constrained Redundant PMs Considering Joint Clearance and Joint Contact Deformation

The offset/deflection of a branch with joint clearance under an external force is affected by multiple factors, such as the clearance size, attitude of mechanism, redundant branch and branch constraint force/couple. Over-constrained redundant PMs generally belong to statically indeterminate structures, and the constraints between branches are complex. Under a certain static attitude and load, the joint with clearance will produce tiny line deflection/angular deflection in its constraint direction under the action of constraint force/torque. It is difficult to judge the joint contact area caused by the branch compound constraint, and it is difficult to establish an accurate mathematical model. In this section, based on certain assumptions, the approximate change interval of the mechanism end is obtained when the joint clearance and joint contact deformation are considered. The specific process is as follows:(1)Firstly, the drive force $f_{a}^{i}$, constraint force $f_{c}^{i}$ and constraint couple $f_{m}^{i}$ of redundant and non-redundant branches are solved by using the system stiffness matrix of the ideal joint.(2)It is assumed that the additional motion of the branch with clearance is in the same direction as the restraint force of the branch. According to the size and direction of the restraint force of each branch, the additional motion direction of the branch with gaps can be determined. The static stiffness modeling process of redundant over-constrained PMs when considering joint clearance is shown in Figure 1.(3)First, the coupling effect of the constraint and constraint couple is ignored and the constraint is considered as a pure force or a pure couple. In most working conditions, the drive force of each branch is much greater than the constraint force, so it is assumed that the clearance joint offset/deflection and joint contact area in the drive force direction reach the maximum. It is also assumed that the ratio of the drive force $f_{a}^{i}$ in the branch with clearance to the offset $\Delta Xi_{f_{a}}^{\mathrm{cle}}$ along the drive force direction and the ratio of the constraint force $f_{c}^{i}$ to the offset $\Delta Xi_{f_{c}}^{\mathrm{cle}}$ along the constraint direction are equal and regarded as constant (which can be approximately equivalent to a spring system with equal stiffness)
$$\frac{f_{a}^{i}}{\Delta Xi_{f_{a}}^{\mathrm{cle}}}=\frac{f_{c}^{i}}{\Delta Xi_{f_{c}}^{\mathrm{cle}}}=k$$

Therefore, the offset of each branch along the constraint direction is
$$\Delta Xi_{f_{c}}^{\mathrm{cle}}=\frac{f_{c}^{i}}{f_{a}^{i}}\Delta Xi_{f_{a}}^{\mathrm{cle}}$$

(4)The offset/deflection of the end of each branch caused by the joint clearance and joint contact deformation is mapped to the deformation of the moving platform by the principle of virtual work. When the deviation of the restraint direction of the clearance joint does not reach the maximum value, the direction also does not reach the contact state and there is no contact deformation between the joints. The static stiffness modeling process of the redundant PM when considering the contact deformation of the joint is shown in Figure 2 ($\Delta Ki_{f_{c}}^{\mathrm{cle}}$ represents the contact deformation caused by the constraint of branch *i*, $\Delta Ki_{f_{a}}^{\mathrm{cle}}$ represents the contact deformation caused by the driving force of branch *i* and $\Delta Ki_{m_{c}}^{\mathrm{cle}}$ represents the angular contact deformation caused by the constraint couple of branch *i*.).(5)In order to solve the uncertainty of the gap joint offset value and joint contact area caused by the branch-coupling force, a random method can be used to describe the variation characteristics of the gap and contact area. $\varsigma_{i}$ is the correction coefficient of random physical variables. Based on the maximum value of the clearance joint offset value and joint contact area, a specific probability distribution attribute is assigned. Using the Monte Carlo method, the probability distribution of the stiffness of the mechanism under a given position and attitude is obtained. The influence of joint clearance and joint contact deformation on the end stiffness is predicted through the probability statistical model, and the variation range of the end stiffness is approximately determined.

## 3. Example Analysis of 2UPR-RR-2RPU Redundant PM Stiffness Considering Multiple Factors

In this paper, a 2UPR-RR-2RPU redundant over-constrained PM was taken as an example to verify the accuracy of the proposed stiffness modeling method. The mechanism is a loop decoupled PM that adopts the method of adding two redundant branches to improve the overall bearing capacity and deformation resistance of the output end.

### 3.1. Configuration Analysis of 2UPR-RR-2RPU Redundant PM

As shown in Figure 3, the 2UPR-RR-2RPU PM consists of two UPR branches, one RR branch and two RPU branches for a total of five branches. The fixed coordinate system $o-x_{0}y_{0}z_{0}$ is established at the center point of the fixed platform. The *x*_0_ axis along the *oA*_4_ direction, the *y*_0_ axis along the *oA*_1_ direction, and the *z*_0_ axis is determined according to the right-hand rule. The moving coordinate system $o_{1}-x_{1}y_{1}z_{1}$ is established at the center point of the fixed platform. The *x*_1_ axis along the *o*_1_*B*_4_ direction, the *y*_0_ axis along the *o*_1_*B*_1_ direction, and the *z*_1_ axis is determined according to the right-hand rule. The middle branch of RR is a just-constrained branch and its axis of rotation connected to the fixed platform is along the direction of $x_{0}$ in the fixed coordinate system $o-x_{0}y_{0}z_{0}$. The revolute pair provides the *x*-direction DOF of rotation of the moving platform around the axis. Its axis of rotation connected to the moving platform is along the direction $y_{1}$ in the moving coordinate system $o_{1}-x_{1}y_{1}z_{1}$. The R pair provides the *y*-direction DOF of rotation of the moving platform around the axis. Among them, the R pairs of the two UPR branches whose respective U pairs are connected to the fixed platform and the R pairs of the RR just-constrained branch that are connected to the fixed platform are coaxial. The respective R pairs of the two UPR branches are parallel to the R pairs of the RR just-constrained branch, which is connected to the moving platform. Among them, the R pair of the two RPU branches whose U pairs are connected to the moving platform is coaxial with the R pair that is connected to the RR just-constrained branch and moving platform, and the respective R pairs of the two RPU branches are, respectively, parallel to the R pairs that are connected to the RR just-constrained branches and fixed platform.

### 3.2. Stiffness Analysis of 2UPR-RR-2RPU Over-Constrained Redundant PM with Ideal Joint

#### 3.2.1. Full Jacobian Matrix Solution for 2UPR-RR-2RPU Redundant PM

In order to obtain the full Jacobian matrix of the mechanism, the constraint Jacobian matrix and the driving Jacobian matrix of the mechanism should be solved first.

$A_{i}B_{i}(i=1,2,3,4)$ represents the direction vector of the kinematics pair of the four driving branches, i=1,2 is the non-redundant branch and i=3,4 is the redundant branch. $A_{i}$ indicates the coordinates of the joint center between each branch and the fixed platform, and $B_{i}$ indicates the center coordinates of the joint that connects each branch and the moving platform.

It can be represented by a screw as
(16)$${\$}_{\mathrm{ai}}={\left(S_{i};\;r_{i}\times S_{i}\right)}^{T}$$

In Equation (16), $S_{i}=A_{i}B_{i}/\left|A_{i}-B_{i}\right|$, $r_{i}$ indicates the vector diameter of each drive branch.
(17)$$\begin{array}{l} J_{\mathrm{na}}^{}={\left[\begin{array}{cc} {\$}_{a1} & {\$}_{a2} \end{array}\right]}^{T}\\ J_{\mathrm{ra}}^{}={\left[\begin{array}{cc} {\$}_{a3} & {\$}_{a4} \end{array}\right]}^{T} \end{array}$$

Let the rotation angle of the mechanism around the rotation axis $x_{o}$ be α and the rotation angle around the rotation axis $y_{1}$ be β, and let a,b be the radius of the fixed platform and the moving platform, respectively.

According to the screw theory, the screw that is reciprocal to the non-redundant UPR branched motion screw is its constraint screw.
(18)$c11f=(0 cosα sinα; 0 −asinα acosα)T$mc21=(0 0 0; 0 −sinα cosα)T

Redundant UPR branches and non-redundant UPR branches are symmetrically arranged in the plane of the base coordinate system with respect to the RR just-constrained branch. In order to obtain the constraint screw $fc13, $mc23 and driving force screw ${\$}_{a3}$ of the redundant UPR branch, it is only necessary to invert the constraint screw $fc11, $mc21 and $a_{1}$, $b_{1}$ in the driving force screw ${\$}_{a1}$ of the non-redundant UPR branch.
(19)$fc11=(0 cosα sinα; 0 asinα −acosα)T$mc21=(0 0 0; 0 −sinα cosα)T

According to the screw theory, the screw that is reciprocal to the non-redundant RPU branched motion screw is its constraint screw
(20)$fc12=(1 0 0; 0 bsinα+dcosα −bcosα+dsinα)T$mc22=(0 0 0; 0 −sinα cosα)T

The redundant RPU branches and the non-redundant RPU branches are symmetrically arranged on the base coordinate system $y_{o}z_{o}$ plane with the RR just-constrained branches. In order to obtain the constraint screw $fc14, $mc24 and driving force screw ${\$}_{a4}$ of the redundant RPU branch, it is only necessary to invert the constraint screw $fc12, $mc22 and $a_{1}$, $b_{1}$ in the driving force screw ${\$}_{a4}$ of the non-redundant UPR branch.
(21)$fc14=1 0 0; 0 −bsinα+dcosα bcosα+dsinαT$mc24=0 0 0; 0 −sinα cosαT

The RR intermediate branch-constrained screw is expressed in the fixed coordinate system as
(22)$fc15=1 0 0; 0 dcosα dsinαT$fc25=0 cosα sinα; 0 0 0T$fc35=0 −sinα cosα; 0 0 0T$mc45=0 0 0; 0 −sinα cosαT

From Equation (3), it can be obtained that the non-redundant constrained Jacobian matrix $J_{\mathrm{nc}}^{}$ and the redundant constrained Jacobian matrix $J_{\mathrm{rc}}^{}$ of the 2UPR-RR-2RPU PM are
(23)Jnc=$fc11 $mc21 $fc12 $mc22 $fc13 $fc23 $fc33 $mc43TJrc=$fc13 $mc23 $fc14 $mc24T

#### 3.2.2. Solution of Driving Stiffness Matrix of UPR-RR-2RPU Redundant PM Branch

The following assumptions are made for the stiffness model: the weight of all components is ignored; all joint models are frictionless; the moving platform is assumed to be a rigid body; and the spatial composite elastic deformation of the branch is considered, including tensile, shear, bending and torsional deformation.

The branch driving stiffness can be simplified as a series spring system, and the driving stiffness coefficient $k_{\mathrm{ai}}$ can be expressed as
(24)$$k_{\mathrm{ai}}^{-1}=\sum_{j=1}^{4}k_{\mathrm{ai},j}^{-1}\;i=1,2,3,4$$

In the Equation, it is expressed as a U pair along the rod axial stiffness $k_{\mathrm{ai},1}^{}$, pendulum rod stiffness $k_{\mathrm{ai},2}^{}$, telescopic rod stiffness $k_{\mathrm{ai},3}^{}$, and R pair seat along the rod axial stiffness $k_{\mathrm{ai},4}^{}$. Among them, $k_{\mathrm{ai},2}^{}$, $k_{\mathrm{ai},3}^{}$ are constants. $k_{\mathrm{ai},2}^{}$ can be calculated by $k_{\mathrm{ai},2}^{}=\mathrm{EA}/\left(q_{i}-l_{1}\right)$, *EA* represents the tensile modulus of the pendulum rod, $l_{1}$ represents the length of the telescopic rod and $q_{i}$ is the distance between the joint center connected with the moving platform and the joint center connected with the fixed platform.

#### 3.2.3. Solution of Branch Restraint Stiffness Matrix of 2UPR-RR-2RPU Redundant PM

The force diagram of the UPR branch is shown in Figure 4. The branch coordinate system $o_{1}-x_{1}y_{1}z_{1}$ is established at the rotation center of the branch close to the moving platform. The $y_{1}$ axis is parallel to the axis direction of the R pair and the $z_{1}$ axis is along the axis of the rod. $x_{1}$ is determined by the right-hand screw rule. In order to analyze the deformation of the branch under the restraint force, the restraint force $f_{11}$ passing through the U pair center and parallel to the R pair direction is translated to $o_{1}-x_{1}y_{1}z_{1}$. According to the force translation theorem, there will be an additional force couple passing through the origin of the branch coordinate system $o_{1}-x_{1}y_{1}z_{1}$ and the direction parallel to the axis $x_{1}$ with magnitude $m_{12}$. There will also be an additional constraint force passing through the origin of the branch coordinate system $o_{1}-x_{1}y_{1}z_{1}$ and the direction parallel to the axis $y_{1}$ with magnitude $m_{11}$. The branch is also constrained by a couple whose axis is parallel to the U pair normal and whose magnitude is $m_{11}$. $f_{1x}$ represents the force in the *x*-axis direction at the cross-section of the RR branch, $f_{1y}$ represents the force in the *y*-axis direction at the cross-section, $f_{1z}$ represents the force in the *z*-axis direction at the cross-section, $m_{1x}$ represents the force couple in the *x*-axis direction at the cross-section, and $m_{1z}$ represents the force couple in the *z*-axis direction at the cross-section.

The internal force of the section where the $l_{i}$ is displaced from the negative direction of the coordinate system $z_{1}$ to the UPR branch can be expressed as
(25)f1x=0f1y=f11f1z=0m1x=m11ςm·e1+m12−f11lim1y=m11ςm·e2=0m1z=m11ςm·e3

In the Equation, ςm is the direction vector of the constraint couple $m_{11}$, $e_{1}$ is the direction vector of the axis $x_{1}$ in the coordinate system $o_{1}-x_{1}y_{1}z_{1}$, $e_{2}$ is the direction vector of the axis $y_{1}$ in the coordinate system $o_{1}-x_{1}y_{1}z_{1}$, $e_{3}$ is the direction vector of the axis $z_{1}$ in the coordinate system $o_{1}-x_{1}y_{1}z_{1}$ and $m_{12}=f_{11}L_{1}$, where $L_{1}$ is the length of the chain rod.

The strain energy of UPR branches can be expressed as
(26)UUPR=∫0li1f1y22Gi1Ai1y+m1x22Ei1Ii1x+m1z22Gi1Ji1dli+∫li1lif1y22Gi2Ai2y+m1x22Ei2Ii2x+m1z22Gi2Ji2dli=∫0li1f1122Gi1Ai1y+m11ςm·e1+m12−f11li22Ei1Ii1x+m11ςm·e322Gi1Ji1dli+∫lili1f1122Gi2Ai2y+m11ςm·e1+m12−f11li22Ei2Ii2x+m11ςm·e322Gi2Ji2dli

In the Equation, $l_{i1}$ is the length of the linear actuator and $L_{1}$ is the total length of the UPR branch and joint center that connect the moving and fixed platforms. When $l_{i}=L_{1}$,
(27)UUPR=li12Gi1Ai1yf112+12Ei1Ii1xMi12li1+13f112li13−Mi1f11li12+12Gi1Ji1m11ςm·e32li1+L1−li12Gi1Ai1yf112+12Ei1Ii1xMi12L1−li1+13f112L13−li13−12Ei1Ii1xMi1f11L12−li12+12Gi1Ji1m11ςm·e32L1−li1

In the Equation,
(28)M11=li1L12−li12L1+13li13ςm·e12Ei1Ii1xM12=L1−li1L12+L12−li12L1+13L13−li13ςm·e12Ei2Ii2xM13=ςm·e12li12Ei1Ii1x,M14=ςm·e12L1−li12Ei2Ii2xM15=2li1L1−li12ςm·e12Ei1Ii1xM16=2L1−li1L1−L12−li12ςm·e12Ei2Ii2x

$E_{i1}$, $E_{i2}$, $G_{i1}$, $G_{i2}$ are the elastic modulus and shear modulus of the telescopic rod and lead screw. $A_{i1y}$, $A_{i2y}$ are the effective shear area along the axis $y_{i}$ of the telescopic rod and the lead screw. $I_{i1x}$, $I_{i2x}$ are the inertia moment of the cross-section of the telescopic rod and the lead screw about the axis $x_{i}$. $J_{i1}$, $J_{i2}$ are the cross-sectional polar inertia moments of the telescopic rod and the lead screw, respectively.

According to Castigliano’s theorem, we can obtain
(29)δ11=∂UUPR∂f11=li1Gi1Ai1y+L1−li1Gi2Ai2y+2M11+2M12f11+M15+M16m11γ11=∂UUPR∂m11=M15+M16f11+2M13+2M14+ςm·e32li1Gi1Ji1+ςm·e32L1−li1Gi2Ji2m11

Arranging the above Equation into matrix form, we can obtain
(30)$$\left[\begin{array}{c} \delta_{11}\\ \gamma_{11} \end{array}\right]=C_{\mathrm{UPR}}\left[\begin{array}{c} f_{11}^{}\\ m_{11}^{} \end{array}\right]$$

In the Equation
(31)CUPR=li1Gi1Ai1y+L1−li1Gi2Ai2y+2M11+2M12M15+M16M15+M162M13+2M14+ςm·e32li1Gi1Ji1+ςm·e32L1−li1Gi2Ji2
the stiffness matrix $K_{\mathrm{UPR}}$ of the UPR branch is the inverse of the compliance matrix $C_{\mathrm{UPR}}$.
(32)$$K_{\mathrm{UPR}}=C_{\mathrm{UPR}}^{-1}$$

It can be seen from the UPR branch compliance matrix that the compliance matrix is not a diagonal matrix. There is a coupling relationship between the constraint force $f_{11}$ and the constraint couple $m_{11}^{}$ and their corresponding elastic deformations. The size of the compliance matrix is related to the direction of the normal vector ςm of the U pair in the UPR branch.

The force diagram of the RPU branch is shown in Figure 5. The branch coordinate system $o_{2}-x_{2}y_{2}z_{2}$ is established at the U pair center of the branch close to the moving platform. The axis $y_{1}$ is parallel to the axis direction of the R pair of the U pair close to the moving platform and the axis $z_{1}$ is along the axis direction of the rod. $x_{1}$ is determined by the right-hand screw rule. According to the screw theory, the RPU branch is subject to a constraint force passing through the origin of the branch coordinate system $o_{2}-x_{2}y_{2}z_{2}$ and the direction parallel to the axis $y_{1}$ with a magnitude of $f_{21}$. It is also subject to a constraint couple whose axis is parallel to the U pair normal with a magnitude of $m_{21}$. $f_{2x}$ represents the force in the *x*-axis direction at the cross-section of the RR branch, $f_{2y}$ represents the force in the *y*-axis direction at the cross-section, $f_{2z}$ represents the force in the *z*-axis direction at the cross-section, $m_{2x}$ represents the force couple in the *x*-axis direction at the cross-section, and $m_{2z}$ represents the force couple in the *z*-axis direction at the cross-section.

The internal force of the section where the displacement from the RPU branch to the negative direction of the branch coordinate system $z_{2}$ is $l_{i}$ can be expressed as
(33)f2x=0f2y=f21f2z=0m2x=m21ξm·e1−f21lim2y=m21ξm·e2=0m2z=m21ξm·e3

In the Equation, ξm is the direction screw of the constraint couple $m_{21}$. $e_{1}$ is the direction vector of the axis $x_{1}$ of the coordinate system $o_{2}-x_{2}y_{2}z_{2}$. $e_{2}$ is the direction vector of the axis $y_{1}$ of the coordinate system $o_{2}-x_{2}y_{2}z_{2}$. $e_{3}$ is the direction vector of the axis $z_{1}$ of the coordinate system $o_{2}-x_{2}y_{2}z_{2}$.

The strain energy of RPU branches can be expressed as
(34)$$\begin{array}{ll} U_{\mathrm{RPU}} & =\int_{0}^{l_{i1}}\left(\frac{f_{2y}^{2}}{2G_{i1}A_{i1y}}+\frac{m_{2x}^{2}}{2E_{i1}I_{i1x}}+\frac{m_{2z}^{2}}{2G_{i1}J_{i1}}\right)dl_{i}\\ & +\int_{l_{i1}}^{l_{i}}\left(\frac{f_{2y}^{2}}{2G_{i2}A_{i2y}}+\frac{m_{2x}^{2}}{2E_{i2}I_{i2x}}+\frac{m_{2z}^{2}}{2G_{i2}J_{i12}}\right)dl_{i} \end{array}$$

In the Equation, $l_{i1}$ is the driving displacement of the driving pair when $l_{i}=l_{2}$:(35)URPU=∫0li1f2y22Gi1Ai1y+m2x22Ei1Ii1x+m2z22Gi1Ji1dli+∫li1lif2y22Gi2Ai2y+m2x22Ei2Ii2x+m2z22Gi2Ji12dli=li12Gi1Ai1y+L2−li12Gi2Ai2y+li136Ei1Ii1x+L23−li136Ei2Ii2xf212+ξm·e12li12Ei1Ii1x+ξm·e32li12Gi1Ji1+ξm·e12L2−li12Ei2Ii2x+ξm·e32L2−li12Gi2Ji2m212−ξm·e1li122Ei1Ii1x+ξm·e1L22−li122Ei2Ii2xf21m21

According to Castigliano’s theorem, we can obtain
(36)$$\begin{array}{c} \delta_{21}=\frac{\partial U_{\mathrm{RPU}}}{\partial f_{21}}=M_{21}f_{21}^{}-M_{22}m_{21}\\ \gamma_{21}=\frac{\partial U_{\mathrm{RPU}}}{\partial m_{21}}=-M_{22}f_{21}+M_{23}m_{21}^{} \end{array}$$

In the Equation,
(37)M21=li1Gi1Ai1y+L2−li1Gi2Ai2y+li133Ei1Ii1x+L23−li133Ei2Ii2xM22=ξm·e1li122Ei1Ii1x+ξm·e1L22−li122Ei2Ii2xM23=ξm·e12li1Ei1Ii1x+ξm·e32li1Gi1Ji1+ξm·e12L2−li1Ei2Ii2x+ξm·e32L2−li1Gi2Ji2

Arranging the above Equation into matrix form, we can obtain
(38)$$\left[\begin{array}{c} \delta_{21}\\ \gamma_{21} \end{array}\right]=C_{\mathrm{RPU}}\left[\begin{array}{c} f_{21}^{}\\ m_{21}^{} \end{array}\right]$$

In the Equation,
(39)$$C_{\mathrm{RPU}}=\left[\begin{array}{cc} M_{21} & -M_{22}\\ -M_{22} & M_{23} \end{array}\right]$$

The stiffness matrix of the RPU branch is the inverse of the compliance matrix
(40)$$K_{\mathrm{RPU}}=C_{\mathrm{RPU}}^{-1}$$

It can be seen from the RPU branch compliance matrix that the compliance matrix is not a diagonal matrix. There is a coupling relationship between the constraint force $f_{21}^{}$, the constraint couple $m_{21}^{}$ and their corresponding elastic deformations. The size of the compliance matrix is related to the direction of the normal vector ξm of the U pair in the RPU branch.

The force figure of the RR just-constrained branch is shown in Figure 6. The branch coordinate system $o_{3}-x_{3}y_{3}z_{3}$ is established at the revolute pair center of the branch close to the moving platform. The axis $y_{1}$ is parallel to the direction of the R pair axis, and the axis $z_{1}$ is along the axis of the rod. $x_{1}$ is determined by the right-hand screw rule. In order to facilitate an analysis of the deformation of the rod under the restraint force, the restraint force $f_{31}$ that passes the R pair center near the fixed platform and is parallel to the R pair axis direction that is close to the moving platform is translated to $o_{3}-x_{3}y_{3}z_{3}$. According to the force translation theorem, there will be an additional force couple passing through the origin of the branch coordinate system $o_{3}-x_{3}y_{3}z_{3}$ and the direction parallel to the axis $x_{3}$ with a magnitude of $m_{32}$. There will also be an additional constraint force passing through the origin of the branch coordinate system $o_{3}-x_{3}y_{3}z_{3}$ and the direction parallel to the axis $y_{3}$ with a magnitude of $f_{31}$. In addition, the branch is also subjected to a constraint couple whose magnitude is $m_{31}$, with the axis parallel to the direction. It is also subjected to a constraint force $f_{33}$ and a constraint force whose axis is parallel to the direction of $y_{3}$ with a magnitude of $f_{32}$. $f_{2x}$ represents the force in the *x*-axis direction at the cross-section of the RR branch, $f_{2y}$ represents the force in the *y*-axis direction at the cross-section, $f_{2z}$ represents the force in the *z*-axis direction at the cross-section, $m_{2x}$ represents the force couple in the *x*-axis direction at the cross-section, and $m_{2z}$ represents the force couple in the *z*-axis direction at the cross-section.

The internal force of the section where the displacement from the RR just-constrained branch to the negative direction of the branch coordinate system $z_{1}$ is $l_{i}$ can be expressed as
(41)$$\left\{\begin{array}{l} f_{3x}=f_{31}\\ f_{3y}=f_{32}\\ f_{3z}=f_{33}\\ m_{3x}=m_{32}-f_{31}l_{i}\\ m_{3y}=f_{32}l_{i}\\ m_{3z}=m_{31} \end{array}\right.$$

In the Equation, $m_{32}=f_{31}L_{3}$, where $L_{3}$ is the rod length of the branch.

The strain energy of the RR-constrained branch can be expressed as
(42)$$\begin{array}{ll} U_{RR} & =\int_{0}^{l_{i}}\frac{f_{3x}^{2}}{2G_{i}A_{ix}}+\frac{f_{3y}^{2}}{2G_{i}A_{iy}}+\frac{f_{3z}^{2}}{2E_{i}A_{i}}+\frac{m_{3x}^{2}}{2E_{i}I_{ix}}+\frac{m_{3y}^{2}}{2E_{i}I_{iy}}+\frac{m_{3z}^{2}}{2G_{i}J_{i}}dl_{i}\\ & =\frac{l_{i}}{2G_{i}A_{ix}}f_{31}^{2}+\frac{l_{i}}{2G_{i}A_{iy}}f_{32}^{2}+\frac{l_{i}}{2E_{i}A_{i}}f_{33}^{2}+\frac{l_{i}}{2E_{i}I_{ix}}m_{32}^{2}+\frac{l_{i}^{3}}{6E_{i}I_{ix}}f_{31}^{2}\\ & -\frac{l_{i}^{2}}{2E_{i}I_{ix}}m_{32}f_{31}+\frac{l_{i}^{3}}{6E_{i}I_{iy}}f_{32}^{2}+\frac{l_{i}}{2G_{i}J_{i}}m_{31}^{2}\\ & =\frac{l_{i}}{2G_{i}A_{ix}}f_{31}^{2}+\frac{l_{i}}{2G_{i}A_{iy}}f_{32}^{2}+\frac{l_{i}}{2E_{i}A_{i}}f_{33}^{2}+\frac{l_{i}^{3}}{6E_{i}I_{ix}}f_{31}^{2}+\frac{l_{i}^{3}}{6E_{i}I_{iy}}f_{32}^{2}\\ & +\frac{l_{i}}{2G_{i}J_{i}}m_{31}^{2} \end{array}$$

According to Castigliano’s theorem, we can obtain
(43)$$\begin{array}{l} \delta_{31}=\frac{\partial U_{RR}}{\partial f_{31}}=\frac{l_{i}}{G_{i}A_{ix}}f_{31}^{}+\frac{l_{i}^{3}}{3E_{i}I_{ix}}f_{31}^{}\\ \delta_{32}=\frac{\partial U_{RR}}{\partial f_{32}}=\frac{l_{i}}{G_{i}A_{iy}}f_{32}^{}+\frac{l_{i}^{3}}{3E_{i}I_{iy}}f_{32}^{}\\ \delta_{33}=\frac{\partial U_{RR}}{\partial f_{33}}=\frac{l_{i}}{E_{i}A_{i}}f_{33}^{}\\ \gamma_{31}=\frac{\partial U_{RR}}{\partial m_{31}}=\frac{l_{i}}{G_{i}J_{i}}m_{31}^{} \end{array}$$

Arranging the above Equation into matrix form, we can obtain
(44)$$\left[\begin{array}{c} \delta_{31}\\ \delta_{32}\\ \delta_{33}\\ \gamma_{31} \end{array}\right]=C_{RR}\left[\begin{array}{c} f_{31}^{}\\ f_{32}^{}\\ f_{33}^{}\\ m_{31}^{} \end{array}\right]$$

In the Equation,
(45)$$C_{RR}=\left[\begin{array}{cccc} \frac{l_{i}}{G_{i}A_{ix}}+\frac{l_{i}^{3}}{3E_{i}I_{ix}} & 0 & 0 & 0\\ 0 & \frac{l_{i}}{G_{i}A_{iy}}+\frac{l_{i}^{3}}{3E_{i}I_{iy}} & 0 & 0\\ 0 & 0 & \frac{l_{i}}{E_{i}A_{i}} & 0\\ 0 & 0 & 0 & \frac{l_{i}}{G_{i}J_{i}} \end{array}\right]$$

The stiffness matrix $K_{RR}$ of the RR just-constrained branch is the inverse matrix of the flexibility matrix $C_{RR}$:(46)$$K_{RR}=C_{RR}^{-1}$$

It can be known from the RR branched compliance matrix that the compliance matrix is a diagonal matrix and that there is no coupling relationship between the constraint force, constraint couple and their corresponding elastic deformation.

## 4. Stiffness Analysis of 2UPR-RR-2RPU Redundant PM Considering Joint Clearance and Joint Contact Deformation

The precise mathematical model of the over-constrained redundant PM with joint clearance is very complicated. The tiny deformation transmitted by each branch with joint clearance to the moving platform is affected by many factors, such as the driving force, posture and attitude, clearance size and external load. The position and attitude of the moving platform is not completely calculated by the inverse kinematic solution, so the moving platform has an active status in the tiny working space, resulting in an uncertain position of the moving platform. An excessive constraint force between joints under large-load conditions will lead to slight penetration deformation, and the penetration deformation of joints will also affect the stiffness of the moving terminal. This cannot be ignored in the field of high precision.

Because most parallel robots have no large deformation during operation, it is assumed that there is no coupling between rigid body displacement and elastic deformation in the calculation. This assumption also ensures that the end stiffness changes caused by joint clearance can be calculated independently.(a)The R pair shaft and the U pair cross shaft are regarded as rigid bodies, and the shaft is in complete contact with the inner wall of the outer cylinder of the rod. (b)The friction force of the joint, the gravity of the rod and the deformation of the branch are ignored. The branch is regarded as a two-force rod.(c)The clearance of the driving pair is not considered.(d)It is assumed that the linear/angular displacement of the joint reaches the maximum value under the action of the constraint force/couple.(e)The clearance at the end of the branch is approximately equal to the linear superposition of the linear/angular displacements of each joint in the branch caused by the constraint force/couple.(f)Without considering the coupling effects of the constraint force and constraint couple on the joint with clearance, the linear displacement/angular displacement of the joint with clearance under the pure force/pure couple is calculated independently, and then a statistical probability model is used to modify the simplified gap stiffness model. 

Usually, the R pair consists of two parts: the rotating shaft and the outer cylinder of the clearance. It is assumed that there is an assembly clearance between the rotating shaft and the rod in the axial and radial directions of the rotating shaft. A local coordinate system o−xyz is established. The axis y is defined as the axis direction of the R auxiliary shaft. The maximum radial clearance is $c_{1}$ and the maximum axial clearance is $c_{2}$. The length of the outer cylinder of the rod is $l_{R}$ and the diameter of the inner wall of the outer cylinder of the rod is $d_{R}$. When the joint is subjected to pure axial and radial constraints, as shown in Figure 7, an offset of the maximum clearance $c_{1}$, $c_{2}$, occurs in the direction of the constraint force.

When the constraint force is the pure constraint couple around the *y*-axis, as shown in Figure 8. This axis is in the direction of rotational degrees of freedom and will not undergo angular deformation; When the constraint force is a pure constraint moment around the *x*-axis or *z*-axis, it can be inferred from the geometric relationship of the clearance model
(47)$$d_{R}-\frac{d_{R}-2c_{1}}{\cos(m_{\theta })}=l_{R}\tan(m_{\theta })$$

Solving the above equation, we can obtain
(48)$$m_{\theta }\approx \frac{-l_{R}+\sqrt{l_{R}^{2}+4d_{R}c_{1}}}{2d_{R}}$$

Without considering the coupling effect of the constraint force and constraint force couple, the simplified approximate clearance model of the R pair can be expressed as
(49)$$\begin{array}{ll} \Delta R^{\mathrm{cle}} & ={\left[\Delta x_{R}^{\mathrm{cle}}\;\Delta y_{R}^{\mathrm{cle}}\;\Delta z_{R}^{\mathrm{cle}}\;\Delta \alpha_{R}^{\mathrm{cle}}\;\Delta \beta_{R}^{\mathrm{cle}}\;\Delta f_{R}^{\mathrm{cle}}\right]}^{T}\\ & ={\left[c_{1}\;c_{2}\;c_{1}\;\left(-l_{R}+\sqrt{l_{R}^{2}+4d_{R}c_{1}}\right)/2d_{R}\;0\;\left(-l_{R}+\sqrt{l_{R}^{2}+4d_{R}c_{1}}\right)/2d_{R}\right]}^{T} \end{array}$$

The UPR branch is subject to the constraint couple $m_{c}$ in the U pair normal direction, the constraint force $f_{c}$ passing through the U pair center and the direction parallel to the R pair axis direction, and the driving force $f_{a}$ along the P pair direction. The deflection of the terminal of the UPR branches when they are individually constrained by forces/couples was analyzed separately.

Figure 9 shows the movement of the U pair with clearance when it is subjected to the driving force along the rod. The clearance model of the U pair can be simplified as a superposition of two revolute pair clearance models with orthogonal axes. The U pair coordinate system is established in Figure 9. Taking the center of the U axis as the coordinate origin and establishing two local coordinate systems $o-u^{\prime }v^{\prime }w^{\prime }$ and o−uvw through the right-hand rule, the axis $u^{\prime }$ and the axis $v^{\prime }$ are along the two orthogonal axis directions of the U axis, respectively, and the coordinate system o−uvw is obtained by rotating the angle α around the axis $v^{\prime }$ of the coordinate system $o-u^{\prime }v^{\prime }w^{\prime }$. The axis w direction of the coordinate system o−uvw is the same as the P pair direction in the UPR branch.

Assuming that the radial and axial clearances of the two orthogonal R pairs in the U pair are $c_{1}$, $c_{2}$ respectively, when the UPR branch is only subjected to the driving force along the direction of the P pair, the rotating shaft where the U pair and the P pair are connected will generate a clearance $c_{1}$ along the radial axis w direction of the rotating shaft and a contact force $f_{u}$ along this direction. The spatial motion of this axis is expressed in matrix form as
(50)$$\Delta U_{R2}^{f_{a}}={\left[c_{2}\;0\;c_{1}\;0\;0\;0\right]}^{T}$$

The first lines of three represent the offsets along the three axes of the coordinate system $o-u^{\prime }v^{\prime }w^{\prime }$, and the last lines of three represent the deflections around the three axes.

Then, the offset/deflection of the U pair in the coordinate system o−uvw under the action of the driving force $f_{a}$ is:(51)ΔUfa=ΔUR1fa+Ro′o0SPoo′Ro′oRo′oΔUR2fa

In the Equation, Ro′o is the rotation transformation matrix of the coordinate system $o-u^{\prime }v^{\prime }w^{\prime }$ relative to the coordinate system o−uvw, and, at this time, SPoo′=0.
(52)Ro′o=cosα0−sinα010sinα0cosα

The offset $\Delta U_{w}^{f_{a}}$ of the UPR branch U pair along the direction of the driving force $f_{a}$ and under the action of the driving force $f_{a}$ can be obtained.
(53)$$\Delta U_{w}^{f_{a}}=c_{1}+c_{2}\sin\alpha +c_{1}\cos\alpha$$

The R pair is acted on by the driving force $f_{a}$, and the offset $\Delta R_{w}^{f_{a}}$ in the direction of the driving force is radial clearance $c_{1}$. 

Therefore, the total offset of the UPR branch in the direction of the driving force produced by the driving force $f_{a}$ is
(54)$$\Delta UPR_{f_{a}}^{\mathrm{cle}}=\Delta U_{w}^{f_{a}}+\Delta R_{w}^{f_{a}}=2c_{1}+c_{2}\sin\alpha +c_{1}\cos\alpha$$

The UPR branch is subjected to a constraint couple along the normal direction of the U axis. The deflection generated by the U pair can be equivalent to the superposition of the deflections of the two R pairs with mutually orthogonal axes around the direction of the constraint couple; that is, $\Delta U_{w^{\prime }}^{m_{c}}=2m_{\theta }$. The deflection generated by the R pair is the deflection $\Delta R_{w^{\prime }}^{m_{c}}$, which is around the direction of the constraint couple, $\Delta R_{w^{\prime }}^{m_{c}}=m_{\theta }$. Then, the total deflection of the UPR branch along the direction of the constraint couple $m_{c}$ produced by the constraint couple $m_{c}$ is
(55)$$\Delta UPR_{f_{c}}^{\mathrm{cle}}=\Delta U_{w}^{f_{c}}+\Delta R_{w}^{f_{c}}=c_{1}+2c_{2}$$

The RPU branch clearance model is the same and will not be introduced here.

The RR intermediate branch clearance model can be equivalent to a U pair, and the branch terminal offset/deflection is
(56)$$\begin{array}{ll} \Delta U^{\mathrm{cle}} & ={\left[\Delta x_{U}^{\mathrm{cle}}\;\Delta y_{U}^{\mathrm{cle}}\;\Delta z_{U}^{\mathrm{cle}}\;\Delta \alpha_{U}^{\mathrm{cle}}\;\Delta \beta_{U}^{\mathrm{cle}}\;\Delta \phi_{U}^{\mathrm{cle}}\right]}^{T}\\ & ={\left[c_{1}+c_{2}\;c_{1}+c_{2}\;2c_{1}\;0\;0\;\left(-l_{R}+\sqrt{l_{R}^{2}+4d_{R}c_{1}}\right)/d_{R}\right]}^{T} \end{array}$$

We can obtain
(57)$$\begin{array}{l} \Delta RR_{f_{c1}}^{\mathrm{cle}}=c_{1}+c_{2}\\ \Delta RR_{f_{c2}}^{\mathrm{cle}}=c_{1}+c_{2}\\ \Delta RR_{f_{c3}}^{\mathrm{cle}}=2c_{1}\\ \Delta RR_{m_{c}}^{\mathrm{cle}}=(-l_{R}+\sqrt{l_{R}^{2}+4d_{R}c_{1}})/d_{R} \end{array}$$

The method of the virtual work principle is suitable for the prediction of the position and attitude accuracy of the mechanism under complex working conditions, and it can achieve a more reasonable description between the joint clearance and the end position and attitude error. The virtual work principle is used to determine the relationship between the position error of each branch and the position error of the platform caused by the clearance. The superposition method is used to quantify the position and attitude error of the branch caused by the clearance of each joint. It is assumed that the offset/deflection of the end of the branch reaches the maximum under the constraint/couple of the branch. It can be obtained from the virtual work principle.
(58)$$\begin{array}{l} {\left(\Delta X^{\mathrm{cle}}\right)}^{T}w=\sum_{i=1}^{2}\Delta \mathrm{UPR}i_{f_{a}}^{\mathrm{cle}}\cdot f_{a}^{\mathrm{RPUi}}+\sum_{i=3}^{4}\Delta \mathrm{RPU}i_{f_{a}}^{\mathrm{cle}}\cdot f_{a}^{\mathrm{UPRi}}\\ +\sum_{i=1}^{2}\left[\begin{array}{cc} \Delta \mathrm{UPR}i_{f_{c}}^{\mathrm{cle}} & \Delta \mathrm{UPR}i_{m_{c}}^{\mathrm{cle}} \end{array}\right]\left[\begin{array}{c} f_{c}^{\mathrm{UPRi}}\\ m_{c}^{\mathrm{UPRi}} \end{array}\right]\\ +\sum_{i=3}^{4}\left[\begin{array}{cc} \Delta \mathrm{RPU}i_{f_{c}}^{\mathrm{cle}} & \Delta \mathrm{RPU}i_{m_{c}}^{\mathrm{cle}} \end{array}\right]\left[\begin{array}{c} f_{c}^{\mathrm{RPUi}}\\ m_{c}^{\mathrm{RPUi}} \end{array}\right]\\ +\left[\begin{array}{cccc} \Delta RR_{f_{c1}}^{\mathrm{cle}} & \Delta RR_{f_{c2}}^{\mathrm{cle}} & \Delta RR_{f_{c3}}^{\mathrm{cle}} & \Delta RR_{m_{c}}^{\mathrm{cle}} \end{array}\right]\left[\begin{array}{c} f_{c1}^{RR}\\ f_{c2}^{RR}\\ f_{c3}^{RR}\\ m_{c}^{RR} \end{array}\right] \end{array}$$

Simplifying further, we can obtain
(59)$$\begin{array}{l} {\left(\Delta X^{\mathrm{cle}}\right)}^{T}w=(K^{\mathrm{cle}}{)}^{T}\tau \\ =\left({\left(K_{\mathrm{na}}^{\mathrm{cle}}\right)}^{T}J_{\mathrm{na}}^{*}+(K_{\mathrm{ra}}^{\mathrm{cle}}{)}^{T}J_{\mathrm{ra}}^{*}+(K_{\mathrm{nc}}^{\mathrm{cle}}{)}^{T}J_{\mathrm{nc}}^{*}+(K_{\mathrm{rc}}^{\mathrm{cle}}{)}^{T}J_{\mathrm{rc}}^{*}\right)w \end{array}$$

In the Equation, $\Delta X^{\mathrm{cle}}$ is the six-DOF deformation of the moving platform caused by the joint clearance.
(60)$$\begin{array}{ll} K_{\mathrm{na}}^{\mathrm{cle}}= & {\left[\Delta \mathrm{UPR}1_{f_{a}}^{\mathrm{cle}}\;\Delta \mathrm{RPU}3_{f_{a}}^{\mathrm{cle}}\right]}^{T}\\ K_{\mathrm{ra}}^{\mathrm{cle}}= & {\left[{\Delta \mathrm{UPR}2}_{f_{a}}^{\mathrm{cle}}\;\Delta \mathrm{RPU}4_{f_{a}}^{\mathrm{cle}}\right]}^{T}\\ K_{\mathrm{nc}}^{\mathrm{cle}}= & [\Delta \mathrm{UPR}1_{f_{c}}^{\mathrm{cle}}\;\Delta \mathrm{UPR}1_{m_{c}}^{\mathrm{cle}}\;\Delta \mathrm{RPU}3_{f_{c}}^{\mathrm{cle}}\;\Delta \mathrm{RPU}3_{m_{c}}^{\mathrm{cle}}\\ & \Delta RR_{f_{c1}}^{\mathrm{cle}}\;\Delta RR_{f_{c2}}^{\mathrm{cle}}\;\Delta RR_{f_{c3}}^{\mathrm{cle}}\;\Delta RR_{m_{c}}^{\mathrm{cle}}{]}^{T}\\ K_{\mathrm{rc}}^{\mathrm{cle}}= & {\left[\Delta \mathrm{UPR}2_{f_{c}}^{\mathrm{cle}}\;\Delta \mathrm{UPR}2_{m_{c}}^{\mathrm{cle}}\;\Delta \mathrm{RPU}4_{f_{c}}^{\mathrm{cle}}\;\Delta \mathrm{RPU}4_{m_{c}}^{\mathrm{cle}}\right]}^{T} \end{array}$$
(61)$$\Delta X^{\mathrm{cle}}=J_{\mathrm{nac}}^{*}K_{\mathrm{nac}}^{\mathrm{cle}}+J_{\mathrm{rac}}^{*}K_{\mathrm{rac}}^{\mathrm{cle}}$$

In the Equation,
(62)$$\begin{array}{l} J_{\mathrm{nac}}^{*}=\mathrm{diag}\left(\begin{array}{cc} {\left(J_{\mathrm{na}}^{*}\right)}^{T} & {\left(J_{\mathrm{nc}}^{*}\right)}^{T} \end{array}\right),J_{\mathrm{rac}}^{*}=\mathrm{diag}\left(\begin{array}{cc} {\left(J_{\mathrm{ra}}^{*}\right)}^{T} & {\left(J_{\mathrm{rc}}^{*}\right)}^{T} \end{array}\right)Z\\ K_{\mathrm{nac}}^{\mathrm{cle}}=\left[\begin{array}{c} K_{\mathrm{na}}^{\mathrm{cle}}\\ K_{\mathrm{nc}}^{\mathrm{cle}} \end{array}\right],K_{\mathrm{rac}}^{\mathrm{cle}}=\left(\begin{array}{c} K_{\mathrm{ra}}^{\mathrm{cle}}\\ K_{\mathrm{rc}}^{\mathrm{cle}} \end{array}\right) \end{array}$$

Due to the coupling effect between the restraint forces, the offsets of the clearance joint along the restraint force direction do not all reach the maximum value. Assuming that the probabilities of the position and attitude of the offset feature elements of all gap joints in the space of the gap joints are equal and meet the normal distribution, the gap joint offset value is given a specific probability distribution attribute on the basis of the maximum value. For the probability distribution of joint clearance, in most cases, it is less than 1. The correction coefficient of random physical variables is defined as the normal distribution coefficient $\varsigma^{N}=(1,1/16)$ whose mean value is 1. Its standard deviation is 1/16 and always meets the left side of the mean value. Therefore, the modified random physical variable considering joint clearance can be expressed as the multiplication of initial physical parameters and random numbers. It includes 16 random variables related to joint clearance. Equation (60) is amended as follows:(63)$$\begin{array}{l} K_{\mathrm{na}}^{\mathrm{cle}}=\varsigma^{N}⊗K_{\mathrm{na}}^{\mathrm{cle}};K_{\mathrm{ra}}^{\mathrm{cle}}=\varsigma^{N}⊗K_{\mathrm{ra}}^{\mathrm{cle}};\\ K_{\mathrm{nc}}^{\mathrm{cle}}=\varsigma^{N}⊗K_{\mathrm{nc}}^{\mathrm{cle}};K_{\mathrm{rc}}^{\mathrm{cle}}=\varsigma^{N}⊗K_{\mathrm{rc}}^{\mathrm{cle}}; \end{array}$$

In the Equation, $C_{m\times N}=a⊗B_{m\times N}$ is defined as multiplying the real number a by each element in the matrix B to obtain the matrix C.

The contact force is approximately equal to the constraint force and is much larger than the friction.

Contact deformation is linear elasticity, regardless of the coefficient of restitution, contact deformation speed and initial impact speed.

The contact force of the joint is approximately equal to the constraint reaction force.

The influence of the coupling of the constraint force and constraint couple on the contact deformation is ignored. The contact deformation under the independent action of the restraint force/moment is calculated separately, and then the calculated results are superposed.

When there is clearance, the restraint force of each branch is approximately equal to that obtained by solving the stiffness matrix of the mechanism when there is no clearance.

There are three cases of the R-pair contact model: (a) plane–plane contact along the axis of the shaft caused by axial restraint; (b) cylinder–cylinder contact along the radial direction of the shaft caused by radial restraint; (c) line–plane contact caused by the radial restraint couple. 

The kinematics pair has two parallel contact surfaces. The contact areas of the contact surface are known. The contact force model is expressed as
(64)$$f_{a}=2E^{*}\sqrt{\frac{A_{a}}{\pi }}d_{a}^{\mathrm{con}}$$

The corrected elastic modulus can be expressed as
(65)$$\frac{1}{E^{*}}=\frac{1-\mu_{1}^{2}}{E_{1}}+\frac{1-\mu_{2}^{2}}{E_{1}}$$

In the Equation, the elastic modulus and Poisson’s ratio of one contact body are $E_{1}$ and $\mu_{1}$, and the elastic modulus and Poisson’s ratio of the other contact body are $E_{2}$ and $\mu_{2}$.

It can be obtained that the contact deformation caused by the axial constraint force is
(66)$$d_{a}^{\mathrm{con}}=k_{a}^{\mathrm{con}}f_{a},\;k_{a}^{\mathrm{con}}=\frac{1}{2E^{*}}\sqrt{\frac{\pi }{A_{a}}}$$

Two cylinders whose axes are parallel to each other are in contact, and the contact force can be expressed as
(67)$$f_{b}=\frac{\pi }{4}E^{*}L_{b}d_{b}^{\mathrm{con}}$$

The contact deformation caused by the radial constraint force is
(68)$$d_{b}^{\mathrm{con}}=k_{b}^{\mathrm{con}}f_{b},k_{b}^{\mathrm{con}}=\frac{4}{\pi E^{*}L_{b}}$$

For the contact deformation caused by the constraint couple in the radial direction, the contact force can be expressed as
(69)$$f_{c}=\frac{\tau_{c}}{L_{c}}$$

The functional relationship between the contact force and the deformation at the contact point is
(70)$$f_{c}=\frac{4}{3}E^{*}R^{\frac{1}{2}}{\left(d_{c}^{\mathrm{con}}\right)}^{\frac{2}{3}}$$

R is the equivalent radius of the two contact bodies, and is shown as follows:(71)$$R=\sqrt{d_{R}\left(d_{R}-2c_{1}\right)}$$
(72)$$\begin{array}{l} q_{c}^{\mathrm{con}}=\frac{d_{c}^{\mathrm{con}}}{L_{c}}=\frac{1}{L_{c}}{\left(\frac{3t_{c}}{4l_{c}E^{*}\sqrt{R}}\right)}^{\frac{2}{3}}=k_{c}^{\mathrm{con}}t_{c}\\ k_{c}^{\mathrm{con}}=\frac{k}{L_{c}}{\left(\frac{3}{4L_{c}E^{*}\sqrt{R}}\right)}^{\frac{2}{3}} \end{array}$$

It can be seen from the above Equation that the functional relationship between the contact angle deformation of the joint and the constraint couple is not linear. In order to simplify the contact deformation model, the constraint couple within a certain range of the contact deformation model is universal. A correction factor $\kappa =\tau_{c}^{1/3}$ is introduced to linearize the function of the contact angle deformation as a function of the constraining couple.

The R pair approximate contact deformation model can be defined as
(73)$$\begin{array}{ll} \Delta R_{}^{\mathrm{con}} & =\left[d_{b}\;d_{a}\;d_{b}\;\theta_{c}\;0\;\theta_{c}\right]\\ & =\mathrm{diag}\left(k_{b}^{\mathrm{con}}\;k_{a}^{\mathrm{con}}\;k_{b}^{\mathrm{con}}\;k_{c}^{\mathrm{con}}\;0\;k_{c}^{\mathrm{con}}\right)\;f_{R}^{c}\\ & =k_{R}^{\mathrm{con}}f_{R}^{c} \end{array}$$

In the Equation, $f_{R}^{c}$ is the six-dimensional constraint matrix of the R pair.

The U pair joint contact deformation model can be equivalent to the U pair joint clearance model, and the approximate contact deformation model in the U-axis coordinate system $o^{\prime }-u^{\prime }v^{\prime }w^{\prime }$ can be defined as
(74)$$\begin{array}{ll} \Delta U_{o^{\prime }}^{\mathrm{con}} & =\left[d_{a}+d_{b}\;d_{a}+d_{b}\;2d_{b}\;0\;0\;2\theta_{c}\right]\\ & =\mathrm{diag}\left(k_{a}^{\mathrm{con}}+k_{b}^{\mathrm{con}}\;k_{a}^{\mathrm{con}}+k_{b}^{\mathrm{con}}\;2k_{b}^{\mathrm{con}}\;0\;0\;2k_{c}^{\mathrm{con}}\right)\;f_{o^{\prime }}^{c}\\ & =k_{U}^{\mathrm{con}}f_{o^{\prime }}^{c} \end{array}$$

In the Equation, $f_{o^{\prime }}^{c}$ is the six-dimensional constraint matrix in the U-axis coordinate system $o^{\prime }-u^{\prime }v^{\prime }w^{\prime }$.

Assuming that each clearance joint of the mechanism reaches a permanent contact state under a static large load, the contact force is approximately equal to the constraint force. Taking the UPR branch as an example, the contact deformation of the driving pair is ignored. Firstly, the contact deformation caused by the driving force $f_{11}$ along the direction of the driving pair is analyzed. The driving pair is regarded as a two-force rod. The R pair and U pair are connected with the rotating shaft of the driving pair to produce radial contact deformation, and the rotating shaft that the U pair and the fixed platform are connected to produce radial contact and axial contact.

In order to analyze the contact deformation of U pair axis 1 in the direction of the driving force $f_{11}$, the constraint force $f_{11}$ is transformed from the branch coordinate system o−uvw to the U axis coordinate system $o^{\prime }-u^{\prime }v^{\prime }w^{\prime }$.
(75)fo′c=Roo′00Roo′foc

In the Equation,
(76)Roo′=cosα0sinα010−sinα0cosα
(77)$$f_{o}^{c}={\left[0\;0\;f_{11}\;0\;0\;0\right]}^{T}$$

According to Equation (77), the six-dimensional contact deformation of the U pair shaft 1 in the coordinate system $o^{\prime }-u^{\prime }v^{\prime }w^{\prime }$ under the action of the driving force $f_{11}$ is
(78)$$\Delta R_{1}^{\mathrm{con}}=k_{R}^{\mathrm{con}}f_{{}_{o^{\prime }}}^{c}$$

The contact deformation $\Delta R_{1}^{\mathrm{con}}$ generated by the U pair axis 1 is transformed into the branched coordinate system o−uvw.
(79)ΔU1con=Ro′o00Ro′oΔR1con

Therefore, the total contact deformation of the U pair in the direction of the driving force $f_{11}$ is the superposition of the deformation $\Delta U_{1}^{\mathrm{con}}(3,1)$ of the shaft 1 in the direction of the driving force $f_{11}$ and the deformation $\Delta U_{2}^{\mathrm{con}}(3,1)$ of the shaft 2 in the direction of the driving force $f_{11}$.
(80)$$\begin{array}{l} \Delta U_{f_{11}}^{\mathrm{con}}=\Delta U_{1}^{\mathrm{con}}(3,1)+\Delta U_{2}^{\mathrm{con}}(3,1)\\ =\Delta U_{1}^{\mathrm{con}}(3,1)+k_{b}^{\mathrm{con}}f_{11}\\ =f_{11}k_{a}^{\mathrm{con}}\sin^{2}\alpha +f_{11}k_{b}^{\mathrm{con}}\cos^{2}\alpha +f_{11}k_{b}^{\mathrm{con}} \end{array}$$

In the Equation, $\Delta U_{1}^{\mathrm{con}}(3,1)$ represents the element in the third row and the first column of the matrix.

When the UPR branch is only subjected to the driving force $f_{11}$, the R pair produces radial contact deformation $\Delta R_{f_{11}}^{\mathrm{con}}$ in the direction of the driving force $f_{11}$
(81)$$\Delta R_{f_{11}}^{\mathrm{con}}=\Delta R_{}^{\mathrm{con}}(3,1)=f_{11}k_{b}^{\mathrm{con}}$$

The contact offset of the UPR branch in the direction of the driving force caused by the driving force $f_{11}$ is
(82)$$\begin{array}{l} \Delta UPR_{f_{11}}^{\mathrm{con}}=\Delta U_{f_{11}}^{\mathrm{con}}+\Delta R_{f_{11}}^{\mathrm{con}}\\ =f_{11}k_{a}^{\mathrm{con}}\sin^{2}\alpha +f_{11}k_{b}^{\mathrm{con}}\cos^{2}\alpha +2f_{11}k_{b}^{\mathrm{con}}\\ =\Delta \mathrm{KUP}R_{f_{11}}^{\mathrm{con}}f_{11} \end{array}$$

The UPR branch is subject to a constrained couple $m_{11}$ along the direction of the U-axis normal. The contact deflection of the U pair produced by the constraint couple $m_{11}$ can be equivalent to the superposition of the angular contact deformation of the two R pairs whose axes are orthogonal to each other around the direction of the constraint couple; that is, $\Delta U_{m_{11}}^{\mathrm{con}}=2\theta_{c}^{\mathrm{con}}$. The contact deflection of the R pair produced by the constraint couple $m_{11}$ is $\Delta R_{m_{11}}^{\mathrm{con}}=\theta_{c}^{\mathrm{con}}$.

Then, the total angular contact deformation of the UPR branch along the direction of the constraint couple $m_{11}$ produced by the constraining couple $m_{11}$ is
(83)$$\Delta UPR_{m_{11}}^{\mathrm{con}}=\Delta U_{m_{11}}^{\mathrm{con}}+\Delta R_{m_{11}}^{\mathrm{con}}=3\theta_{c}^{\mathrm{con}}=\Delta \mathrm{KUP}R_{m_{11}}^{\mathrm{con}}m_{11}$$

The UPR branch is subjected to a constraint force $f_{12}$ passing through the U pair center and in a direction parallel to the R pair axis. Among them, the U pair produces the contact deformation $d_{a}^{\mathrm{con}}$ in the axial direction of the rotating shaft parallel to the R pair and the contact deformation $d_{b}^{\mathrm{con}}$ in the radial direction of the rotating shaft orthogonal to the R pair. The R pair produces contact deformation $d_{a}^{\mathrm{con}}$ in the axial direction. Then, the contact offset of the UPR branch along the direction of the constraint force $f_{12}$ produced by the constraint force $f_{12}$ is
(84)$$\begin{array}{l} \Delta UPR_{f_{12}}^{\mathrm{con}}=\Delta U_{f_{12}}^{\mathrm{con}}+\Delta R_{f_{12}}^{\mathrm{con}}\\ =2d_{a}^{\mathrm{con}}+d_{b}^{\mathrm{con}}=\Delta \mathrm{KUP}R_{f_{12}}^{\mathrm{con}}f_{12} \end{array}$$

The RPU branch clearance model is the same and will not be introduced here.

The contact model of the RR intermediate branch is equivalently analyzed as the U pair, and it can be obtained that
(85)$$\begin{array}{l} \Delta \mathrm{KR}R_{f_{c1}}^{\mathrm{con}}=k_{a}^{\mathrm{con}}+k_{b}^{\mathrm{con}}\\ \Delta \mathrm{KR}R_{f_{c2}}^{\mathrm{con}}=k_{a}^{\mathrm{con}}+k_{b}^{\mathrm{con}}\\ \Delta \mathrm{KR}R_{f_{c3}}^{\mathrm{con}}=2k_{b}^{\mathrm{con}}\\ \Delta \mathrm{KR}R_{m_{c}}^{\mathrm{con}}=2k_{c}^{\mathrm{con}} \end{array}$$

The terminal 6-DOF directional deformation is quantified as the joint contact deformation and the load function acting on the platform by the principle of virtual work.
(86)$$\begin{array}{l} {\left(\Delta X^{con}\right)}^{T}w=\sum_{i=1}^{2}\Delta UPRi_{f_{a}}^{con}\cdot f_{a}^{RPUi}+\sum_{i=3}^{4}\Delta RPUi_{f_{a}}^{con}\cdot f_{a}^{UPRi}\\ +\sum_{i=1}^{2}\left[\begin{array}{cc} \Delta UPRi_{f_{c}}^{con} & \Delta UPRi_{m_{c}}^{con} \end{array}\right]\left[\begin{array}{c} f_{c}^{UPRi}\\ m_{c}^{UPRi} \end{array}\right]\\ +\sum_{i=3}^{4}\left[\begin{array}{cc} \Delta RPUi_{f_{c}}^{con} & \Delta RPUi_{m_{c}}^{con} \end{array}\right]\left[\begin{array}{c} f_{c}^{RPUi}\\ m_{c}^{RPUi} \end{array}\right]\\ +\left[\begin{array}{cccc} \Delta RR_{f_{c1}}^{con} & \Delta RR_{f_{c2}}^{con} & \Delta RR_{f_{c3}}^{con} & \Delta RR_{m_{c}}^{con} \end{array}\right]\left[\begin{array}{c} f_{c1}^{RR}\\ f_{c2}^{RR}\\ f_{c3}^{RR}\\ m_{c}^{RR} \end{array}\right] \end{array}$$

Simplifying further, we can obtain
(87)$$\begin{array}{l} {\left(\Delta X^{\mathrm{con}}\right)}^{T}w=(K^{\mathrm{con}}\left|\tau \right|{)}^{T}\tau \\ =\left(\begin{array}{l} {\left(K_{\mathrm{na}}^{\mathrm{con}}\left|\tau_{\mathrm{na}}\right|\right)}^{T}J_{\mathrm{na}}^{*}+(K_{\mathrm{ra}}^{\mathrm{con}}\left|\tau_{\mathrm{ra}}\right|{)}^{T}J_{\mathrm{ra}}^{*}\\ +(K_{\mathrm{nc}}^{\mathrm{con}}\left|\tau_{\mathrm{ra}}\right|{)}^{T}J_{\mathrm{nc}}^{*}+(K_{\mathrm{rc}}^{\mathrm{con}}\left|\tau_{\mathrm{ra}}\right|{)}^{T}J_{\mathrm{rc}}^{*} \end{array}\right)w \end{array}$$

In the Equation,
(88)$$\begin{array}{ll} K_{\mathrm{na}}^{\mathrm{con}}= & \mathrm{diag}(\Delta \mathrm{KUPR}1_{f_{a}}^{\mathrm{con}}\;\Delta \mathrm{KRPU}3_{f_{a}}^{\mathrm{con}})\\ K_{\mathrm{ra}}^{\mathrm{con}}= & \mathrm{diag}(\Delta \mathrm{KUPR}2_{f_{a}}^{\mathrm{con}}\;\Delta \mathrm{KRPU}4_{f_{a}}^{\mathrm{con}})\\ K_{\mathrm{nc}}^{\mathrm{con}}= & \mathrm{diag}(\Delta \mathrm{KUPR}1_{f_{c}}^{\mathrm{con}}\;\Delta \mathrm{KUPR}1_{m_{c}}^{\mathrm{con}}\;\Delta \mathrm{KRPU}3_{f_{c}}^{\mathrm{con}}\\ & \Delta \mathrm{KRPU}3_{m_{c}}^{\mathrm{con}}\;\Delta \mathrm{KR}R_{f_{c1}}^{\mathrm{con}}\;\Delta \mathrm{KR}R_{f_{c2}}^{\mathrm{con}}\;\Delta \mathrm{KR}R_{f_{c3}}^{\mathrm{con}}\;\Delta \mathrm{KR}R_{m_{c}}^{\mathrm{con}})\\ K_{\mathrm{rc}}^{\mathrm{con}}= & \mathrm{diag}(\Delta \mathrm{KUPR}2_{f_{c}}^{\mathrm{con}}\;\Delta \mathrm{KUPR}2_{m_{c}}^{\mathrm{con}}\;\Delta \mathrm{KRPU}4_{f_{c}}^{\mathrm{con}}\;\Delta \mathrm{KRPU}4_{m_{c}}^{\mathrm{con}}) \end{array}$$
(89)$$\Delta X^{\mathrm{con}}=K^{\mathrm{con}}\left|\tau \right|=J_{\mathrm{nac}}^{*}K_{\mathrm{nac}}^{\mathrm{con}}\tau_{\mathrm{nac}}+J_{\mathrm{rac}}^{*}K_{\mathrm{rac}}^{\mathrm{con}}\tau_{\mathrm{rac}}$$

In the Equation,
(90)$$\begin{array}{l} K_{\mathrm{nac}}^{\mathrm{con}}=\mathrm{diag}\left(\begin{array}{cc} K_{\mathrm{na}}^{\mathrm{con}} & K_{\mathrm{nc}}^{\mathrm{con}} \end{array}\right),K_{\mathrm{rac}}^{\mathrm{con}}\mathrm{diag}\left(\begin{array}{cc} K_{\mathrm{ra}}^{\mathrm{con}} & K_{\mathrm{rc}}^{\mathrm{con}} \end{array}\right)\\ \tau_{\mathrm{nac}}=\left[\begin{array}{c} \left|\tau_{\mathrm{na}}\right|\\ \left|\tau_{\mathrm{nc}}\right| \end{array}\right],\tau_{\mathrm{rac}}=\left[\begin{array}{c} \left|\tau_{\mathrm{ra}}\right|\\ \left|\tau_{\mathrm{rc}}\right| \end{array}\right] \end{array}$$

Because of the coupling effect between the restraint forces, the contact areas of the joint along the restraint force direction do not all reach the maximum value. This will cause a stiffness modeling error. In order to achieve the high-precision stiffness modeling of the mechanism, it is assumed that all the characteristic elements of the joint contact area are located in all the contact areas within the joint with equal probability and meet the normal distribution. Therefore, the joint contact area is given a specific probability distribution attribute on the basis of the maximum value. For the probability distribution of the joint contact area, in most cases, it is distributed at around less than 1. The correction coefficient $\varsigma^{N}=(1,1/16)$ of random physical variables is defined as the normal distribution coefficient whose mean value is 1. The standard deviation is 1/16 and always meets the left side of the mean value. Therefore, the modified random physical variable considering joint contact deformation can be expressed as the multiplication of initial physical parameters and random numbers. It includes 16 random variables related to joint contact. The correction of Equation (90) is as follows:(91)$$\begin{array}{l} K_{\mathrm{na}}^{\mathrm{con}}=\varsigma^{N}⊗K_{\mathrm{na}}^{\mathrm{con}};K_{\mathrm{ra}}^{\mathrm{con}}=\varsigma^{N}⊗K_{\mathrm{ra}}^{\mathrm{con}};\\ K_{\mathrm{nc}}^{\mathrm{con}}=\varsigma^{N}⊗K_{\mathrm{nc}}^{\mathrm{con}};K_{\mathrm{rc}}^{\mathrm{con}}=\varsigma^{N}⊗K_{\mathrm{rc}}^{\mathrm{con}}; \end{array}$$

## 5. Synthesis Example of Weakly Coupled Three-Translation PM

### 5.1. Global Stiffness Analysis of Ideal Joints

In order to reflect the end deformation of a 2UPR-RR-2RPU redundant PM when driving stiffness and branch deformation are considered, the square root of linear deformation in three directions is taken as the stiffness evaluation index.
(92)$$\left\{\begin{array}{c} \kappa_{d}^{\varsigma }=\sqrt{{\left(\Delta X_{\mathrm{dx}}^{\varsigma }\right)}^{2}+{\left(\Delta X_{\mathrm{dz}}^{\varsigma }\right)}^{2}+{\left(\Delta X_{\mathrm{dy}}^{\varsigma }\right)}^{2}}\\ \kappa_{\varphi }^{\varsigma }=\sqrt{{\left(\Delta X_{\varphi x}^{\varsigma }\right)}^{2}+{\left(\Delta X_{\varphi y}^{\varsigma }\right)}^{2}+{\left(\Delta X_{\varphi z}^{\varsigma }\right)}^{2}} \end{array}\right.$$

In the Equation, $\Delta X_{\mathrm{dx}}^{\varsigma },\Delta X_{\mathrm{dy}}^{\varsigma }$ and $\Delta X_{\mathrm{dz}}^{\varsigma }$ represent the linear deformation along the *x* axis, *y* axis and *z* axis, respectively.$\Delta X_{\varphi x}^{\varsigma },\Delta X_{\varphi y}^{\varsigma }$ and $\Delta X_{\varphi z}^{\varsigma }$ represent the angular deformation along the *x*, *y* and *z* axes, respectively.

The structural parameters of the 2UPR-RR-2RPU redundant PM are as follows: the external diameter of the swing rod is 40 mm; the internal diameter of the swing rod and the external diameter of the push rod are 25 mm; the length of the swing rod is 205 mm; the diameter of the intermediate restraining chain is 45 mm and its height is 320 mm; the radius of the fixed platform is 260 mm; the radius of the moving platform is 210 mm; the elastic modulus is 200 GPa; and the Poisson’s ratio is 0.3. In the analysis of joint clearance, it is assumed that the axial clearance of the joint is 0.1 mm and the radial clearance is 0.05 mm. In the analysis of joint contact deformation, the length of the joint outer cylinder is 105 mm, and the diameter of the joint outer cylinder inner wall is 25 mm given a 7500 N external load downward along the normal direction of the moving platform. This parallel mechanism is applied in the field of wheel coupling fatigue durability testing. The mechanism is connected in series with the *z*-direction actuator cylinder to simulate the force acting on tires on rough, potholed and inclined roads. It compensates for the deficiency of a four-channel road simulation platform that can only provide *z*-direction excitation. The quarter vehicle load is 7500. Through the above stiffness performance evaluation indicators, the global performance distribution of a 2UPR-RR-2RPU redundant PM under four factors was studied, respectively, as shown in Figure 10.

Through the defined stiffness index in this paper, the following conclusions can be drawn.

After the introduction of redundant branches, the stiffness characteristic distribution of the PM is improved. The structural stiffness has good symmetry in the workspace. The linear stiffness and angular stiffness are obvious. The stiffness of the mechanism is less affected by the rotation angle β of the moving platform and the platform is more stable when the attitude angle is output. 

### 5.2. Comparison between Theoretical Model and Finite Element Model of Static Stiffness for Ideal Joint

In order to verify the correctness of the theoretical model under the ideal joint, the Solidworks model was imported into the Ansys Workbench. The material performance parameters were defined as follows: density $\rho =7850\;\mathrm{kg}/m^{3}$, Poisson’s ratio μ=0.3, elastic modulus $E=2\times 10^{11}\;\mathrm{Pa}$. The theoretical model and finite element model of the mechanism under two positions and attitudes were established for comparison. The first attitude: α=0, β=0; The second attitude: α=15°,β=15°. The joint was set as an ideal kinematics pair, and fixed constraints were added to the fixed platform. The moving platform was set as rigid, and the following six-dimensional load was applied at the center point of the moving platform coordinate system: (93)$$w={\left[750\;N\;750\;N\;-750\;N\;750\;N\cdot m\;750\;N\cdot m\;750\;N\cdot m\right]}^{T}$$

The deformation cloud diagram obtained by finite element simulation is shown in Figure 11, and the comparison between theoretical and simulation results is shown in Table 1.

### 5.3. Model Correction Considering Hinge Clearance and Hinge Contact Deformation Stiffness

The given load of the moving platform is as follows:(94)$$w={\left[750\;N\;750\;N\;-7500\;N\;0\;N\cdot m\;0\;N\cdot m\;0\;N\cdot m\right]}^{T}$$

The revised stiffness model considering the clearance and joint contact deformation was calculated 5000 times. The 32 random variables involved in the calculation each time meet the normal distribution. A histogram was drawn and the distribution was fitted for the 5000 groups of final end deformations—that is, the distribution shape of the end stiffness—taking 50 sample spaces within the maximum and minimum range of linear deformation/angular deformation at the end of the 2UPR-RR-2RPU redundant PM. The stiffness distribution of the initial position and attitude considering joint clearance and joint contact deformation is shown in Figure 12 and Figure 13.

According to the stiffness distribution diagram, the following conclusions are drawn.

The fluctuation range and probability distribution of the output end stiffness caused by the random parameters related to 16 joint gaps and 16 joint contacts are clearly shown. According to the law of large numbers, although the linear stiffness/angular stiffness distribution in each direction is slightly different, it basically conforms to the normal distribution.

The shape of the stiffness distribution is determined by the predetermined pose and given parameters (structure size, clearance value, contact parameters, load and variance).

### 5.4. Comparison between Theoretical Model and Finite Element Model Considering Joint Clearance and Joint Contact Deformation

In order to facilitate the convergence of the model, the calculation time was saved. In the simplified model, the connecting rod was set as a rigid body, and the clearance joint was set as frictionless contact. The mesh of the contact area was refined, the contact parameters and iteration steps were set reasonably and the fixed constraints were added to the fixed platform. The deformation of the moving platform with clearance joints under two different loads in the attitude α=0, β=0 is discussed. 

Load 1:(95)$$w={\left[750\;N\;750\;N\;-7500\;N\;0\;N\cdot m\;0\;N\cdot m\;0\;N\cdot m\right]}^{T}$$

Load 2:(96)$$w={\left[750\;N\;750\;N\;-750\;N\;0\;N\cdot m\;0\;N\cdot m\;0\;N\cdot m\right]}^{T}$$

The above load on the moving platform was applied and the contact parameters of the clearance joint (U pair, R pair) were set. The angular deformation around the *x*, *y*, *z* axis tends to zero under the given boundary conditions. Therefore, only the line deformation along the *x*, *y*, *z* direction was analyzed. When the standard deviation is 1/16, the finite element simulation of the stiffness prediction interval at the output end of the attitude and the center of the moving platform is obtained. The simulation results are shown in Table 2, and the total deformation cloud diagram is shown in Figure 14. The deformation in each direction under two different loads is shown in Figure 15 and Figure 16.

As shown in Table 2, when considering the joint clearance and joint contact deformation, the finite element simulation results under the given standard deviation are within the stiffness variation range of the theoretical model. The maximum error of theory and simulation is 5.79% when measuring the error by sample mean. The correctness of the theoretical model was verified through a comparative analysis. The model can be applied to the static stiffness performance evaluation of a 2UPR-RR-2RPU redundant PM at the design stage, laying the foundation for stiffness optimization.

### 5.5. Stiffness Model Analysis of 2UPR-RR-2RPU Redundant PM Considering Multiple Factors

In order to evaluate the accuracy of the joint clearance model and joint contact deformation model under different external loads, the deformation of the end under two different external loads is discussed. The force amplitude of load 1 along the *z* direction is far greater than that along the *x*, *y* directions, whereas the force amplitude of load 2 along the *x*, *y*, *z* directions is equal. Taking *t* as the step size, the simulation time is given as 2 s, and the output motion law and applied dynamic load of the PM are as follows:

Plan attitude and load 1:(97)α=10cos(2πt)⋅π/180β=10sin(2πt)⋅π/180,Fo1x=750cos(2πt) Fo1y=750cos(2πt)Fo1z=−7.5×103

Plan attitude and load 2:(98)α=10cos(2πt)⋅π/180β=10sin(2πt)⋅π/180,Fo1x=750cos(2πt) Fo1y=750cos(2πt)Fo1z=−750

Based on the analysis method proposed in this paper, the driving stiffness, bar deformation, joint clearance and joint contact deformation were considered. Using Matlab software to calculate the 2UPR-RR-2RPU redundant PM, the deformation of the PM output end in the six-DOF direction is caused by various factors under the above load conditions and given motion laws. Statistical analysis was carried out on the modified clearance model and joint contact deformation model. The stiffness performance considering clearance and joint contact deformation was evaluated by means of sample mean and the quality of the statistical simulation results was evaluated. The maximum/minimum values of the samples were used to judge the stiffness fluctuation range considering the clearance and joint contact deformation. The approximate linear/angular stiffness interval of the mechanism in each direction can be determined. The maximum, minimum and average values of line deformation/angular deformation in each direction considering different factors under two different planning attitudes and loads are shown in Figure 17 and Figure 18.

## 6. Conclusions

Firstly, a high-precision stiffness modeling method of an over-constrained redundant PM was proposed. Taking a 2UPR-RR-2RPU over-constrained redundant PM as an example, the ideal joint was derived by using strain energy and Castigliano’s second theorem. The stiffness matrix of the mechanism with its driving stiffness and branch deformation and the restraint force of each branch should be considered.

Based on the principle of virtual work, a stiffness modeling method considering the joint clearance and joint contact deformation was proposed. The probabilistic model was used to predict the position distribution of the clearance joints, and the problem of the modeling error caused by coupling constraints of over-constrained mechanisms was solved. The stiffness variation interval considering the joint clearance and joint contact deformation under different external loads was obtained, and the influence degree of each factor (driving stiffness, bar deformation, joint clearance, joint contact deformation) on the end deformation was analyzed.

Finally, the correctness of the proposed stiffness model was verified using the finite element method. This method can help robot designers and manufacturing engineers to fully consider the influence of multiple factors (driving stiffness, bar deformation, joint clearance, joint contact deformation) on the static stiffness of the mechanism. Through this method, a high-precision static stiffness model can be established to achieve the best performance of the mechanism.

## Figures and Tables

**Figure 1 sensors-23-05916-f001:**
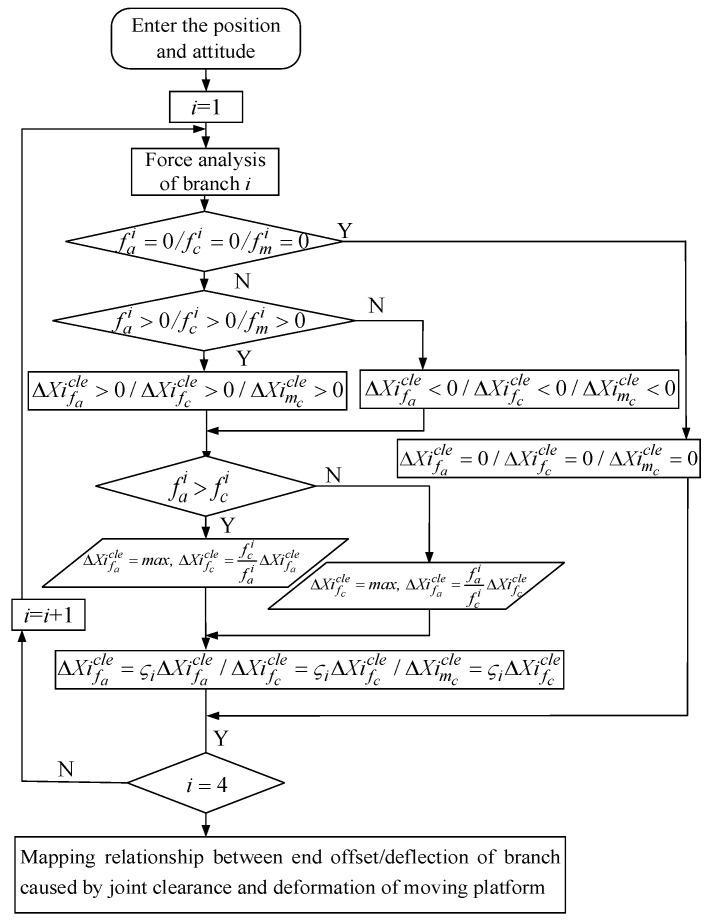
Modeling process of static stiffness of the platform considering joint clearance.

**Figure 2 sensors-23-05916-f002:**
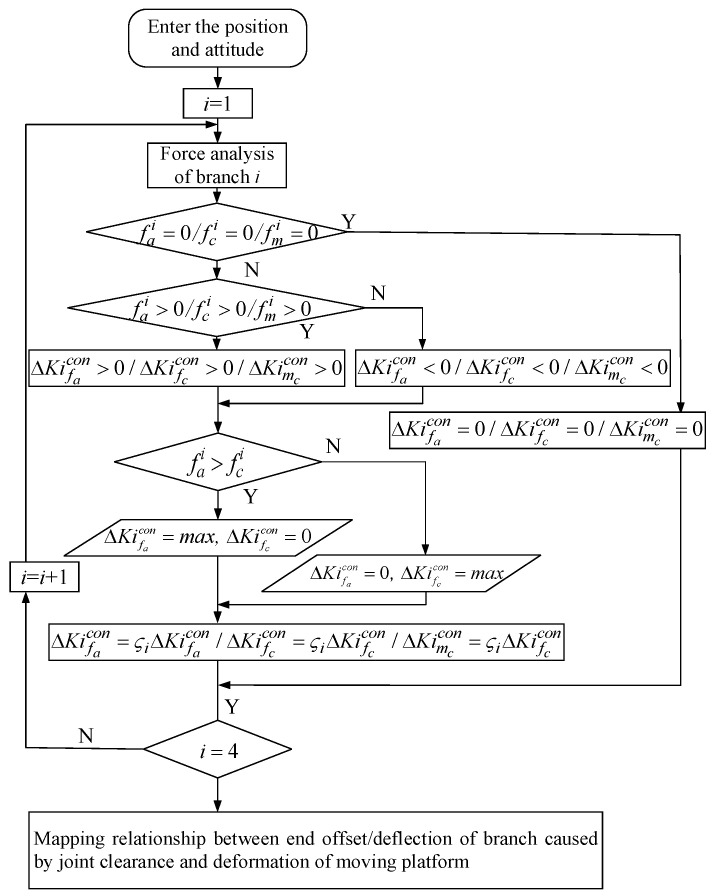
Modeling process of static stiffness of the platform considering joint contact deformation.

**Figure 3 sensors-23-05916-f003:**
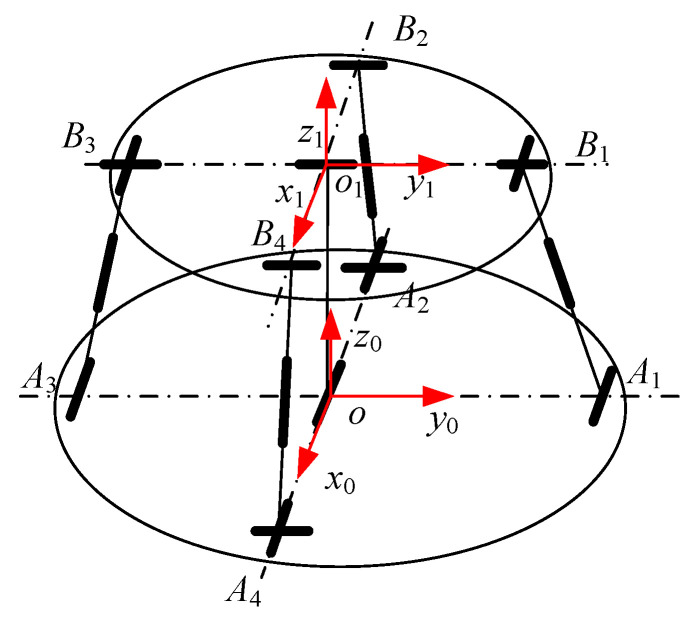
Structural diagram of 2UPR-RR-2RPU redundant PM.

**Figure 4 sensors-23-05916-f004:**
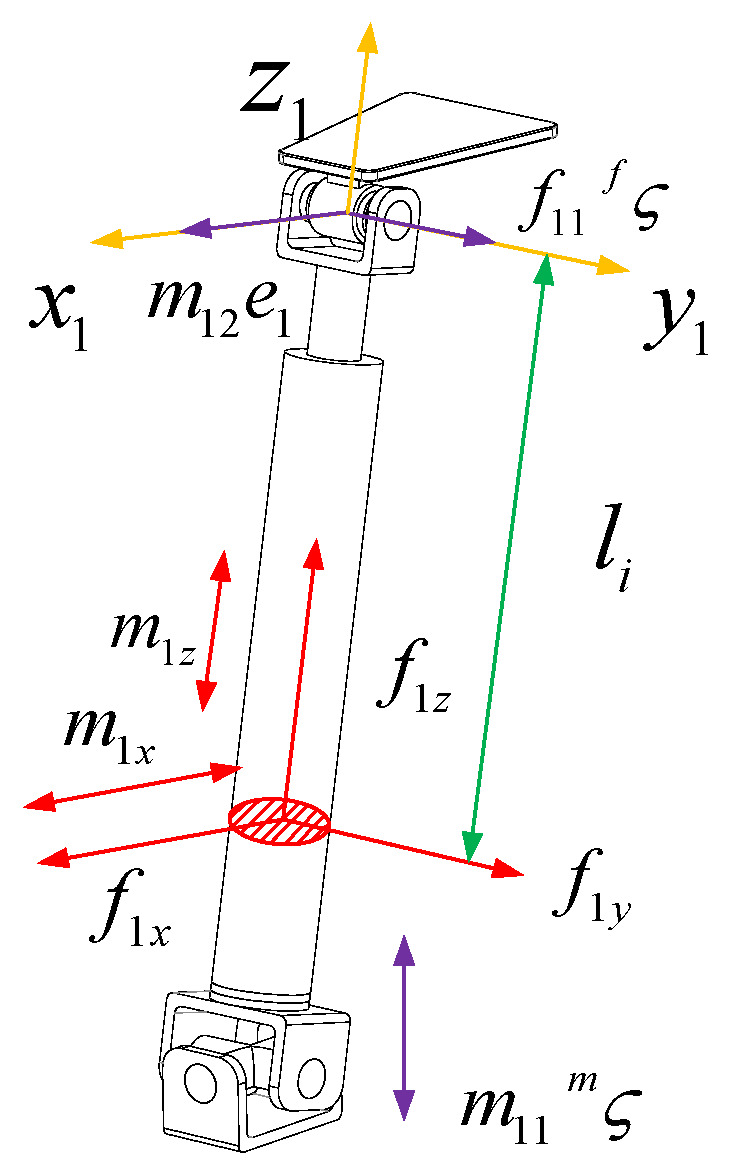
The force figure of the UPR branch chain.

**Figure 5 sensors-23-05916-f005:**
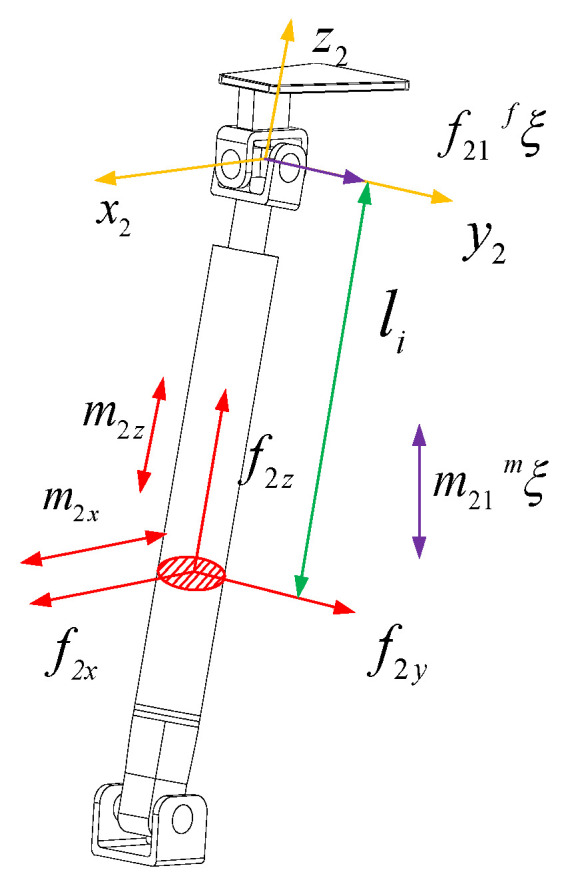
The force figure of the RPU branch.

**Figure 6 sensors-23-05916-f006:**
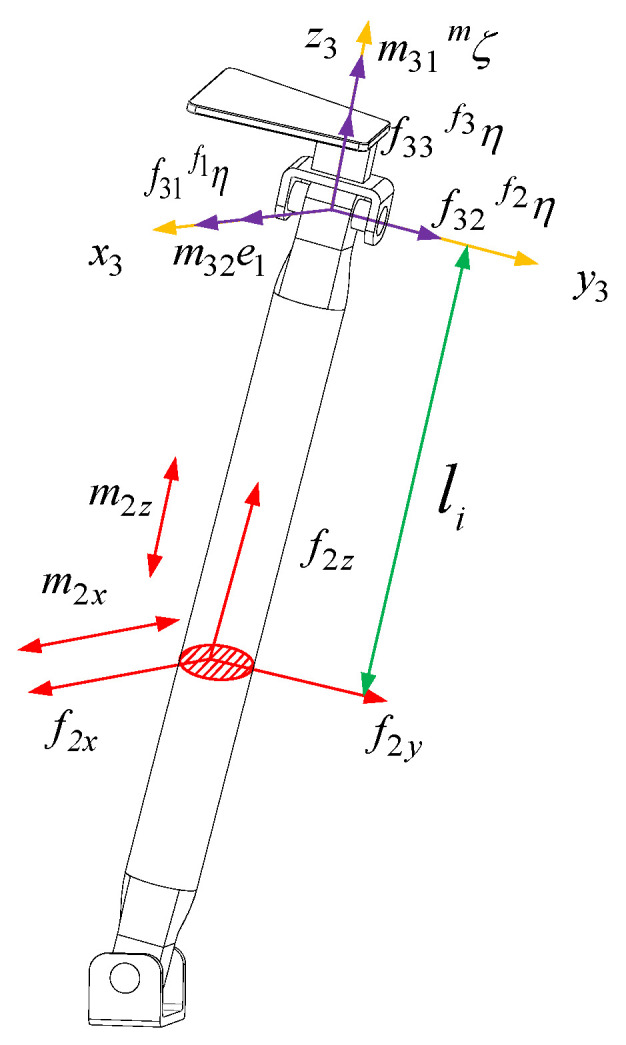
The force figure of the RR-constrained branch chain.

**Figure 7 sensors-23-05916-f007:**
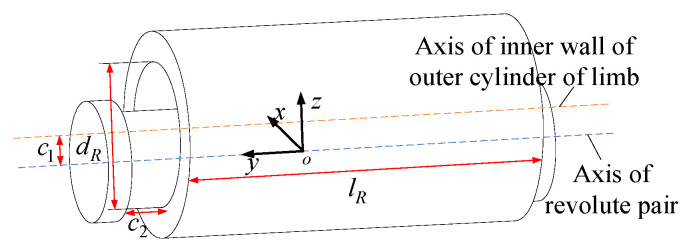
The offset diagram of R pair with clearance under pure constraint force.

**Figure 8 sensors-23-05916-f008:**
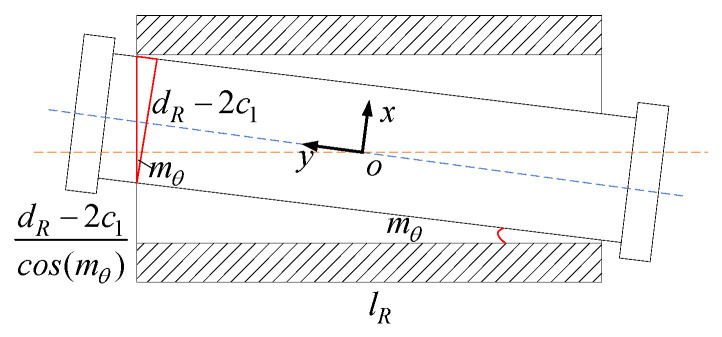
The offset diagram of R pair with clearance under constraint force couple.

**Figure 9 sensors-23-05916-f009:**
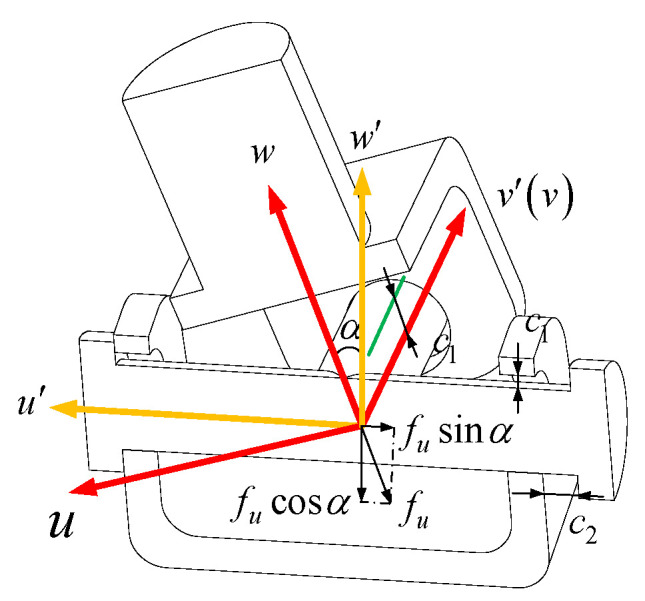
The offset diagram of U pair with clearance under constraint force/moment.

**Figure 10 sensors-23-05916-f010:**
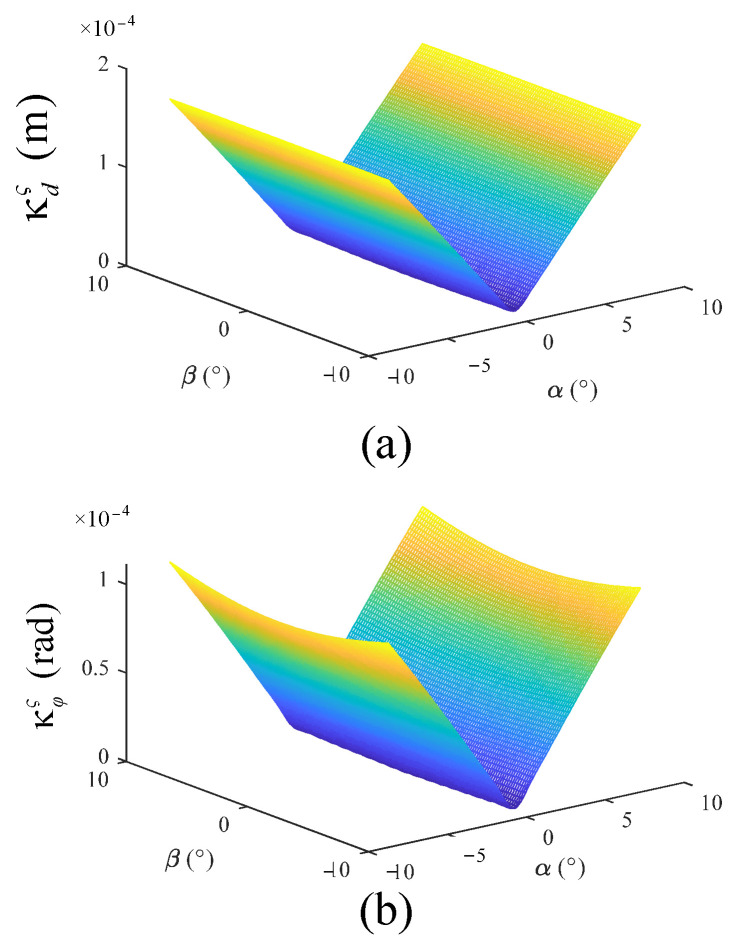
Stiffness distribution diagram of 2UPR-RR-2RPU redundant PM considering different factors. (**a**) Global distribution of linear deformation when driving stiffness and branch deformation are considered; (**b**) global distribution of angular deformation when driving stiffness and branch deformation are considered.

**Figure 11 sensors-23-05916-f011:**
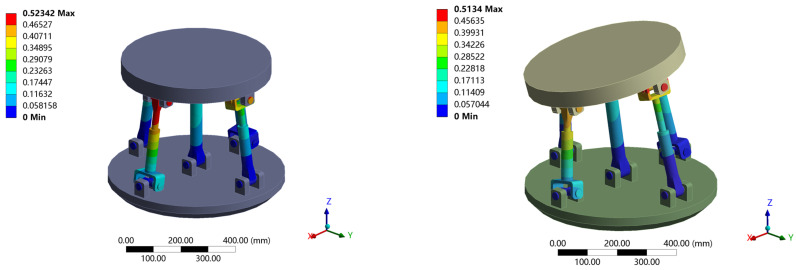
Deformation cloud diagram of 2UPR-RR-2RPU redundant PM with ideal joint.

**Figure 12 sensors-23-05916-f012:**
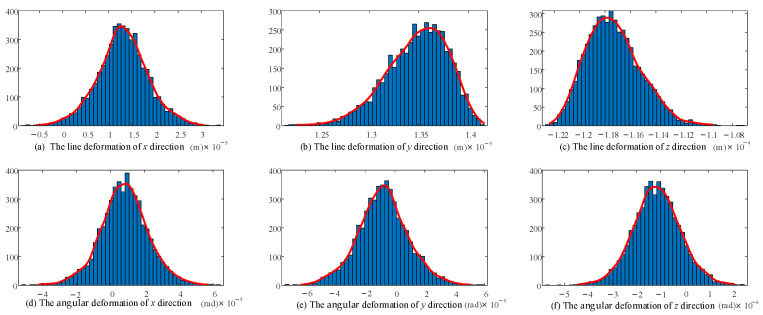
Probability distribution diagram of linear deformation and angular deformation in each direction of the end considering joint clearance.

**Figure 13 sensors-23-05916-f013:**
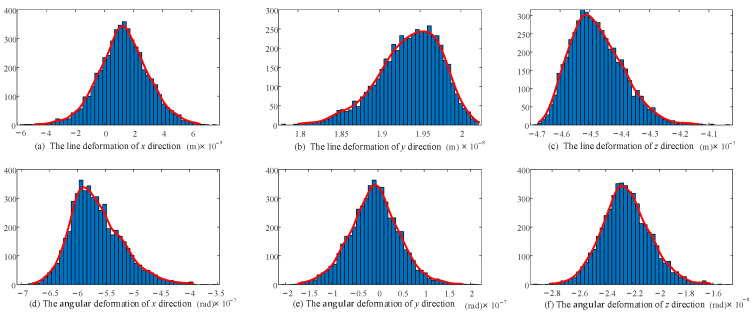
Probability distribution diagram of linear deformation and angular deformation in each direction of the end considering joint contact deformation.

**Figure 14 sensors-23-05916-f014:**
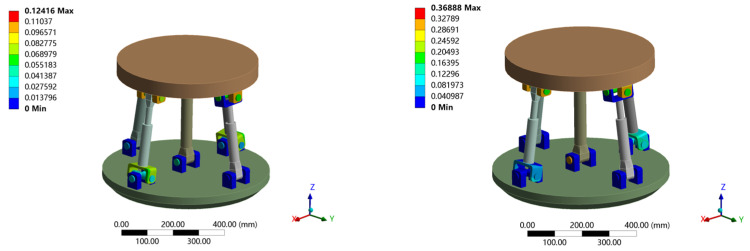
Deformation cloud diagram of 2UPR-RR-2RPU redundant PM considering joint clearance and contact deformation.

**Figure 15 sensors-23-05916-f015:**
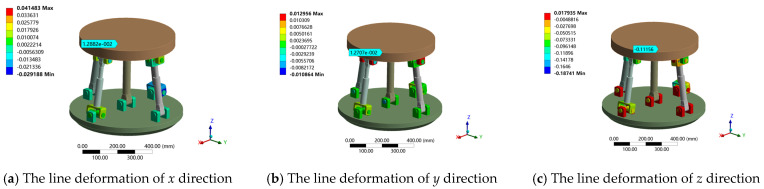
Linear deformation of 2UPR-RR-2RPU redundant PM in *x*, *y* and *z* directions under load 1.

**Figure 16 sensors-23-05916-f016:**

Linear deformation of 2UPR-RR-2RPU redundant PM in *x*, *y* and *z* directions under load 2.

**Figure 17 sensors-23-05916-f017:**
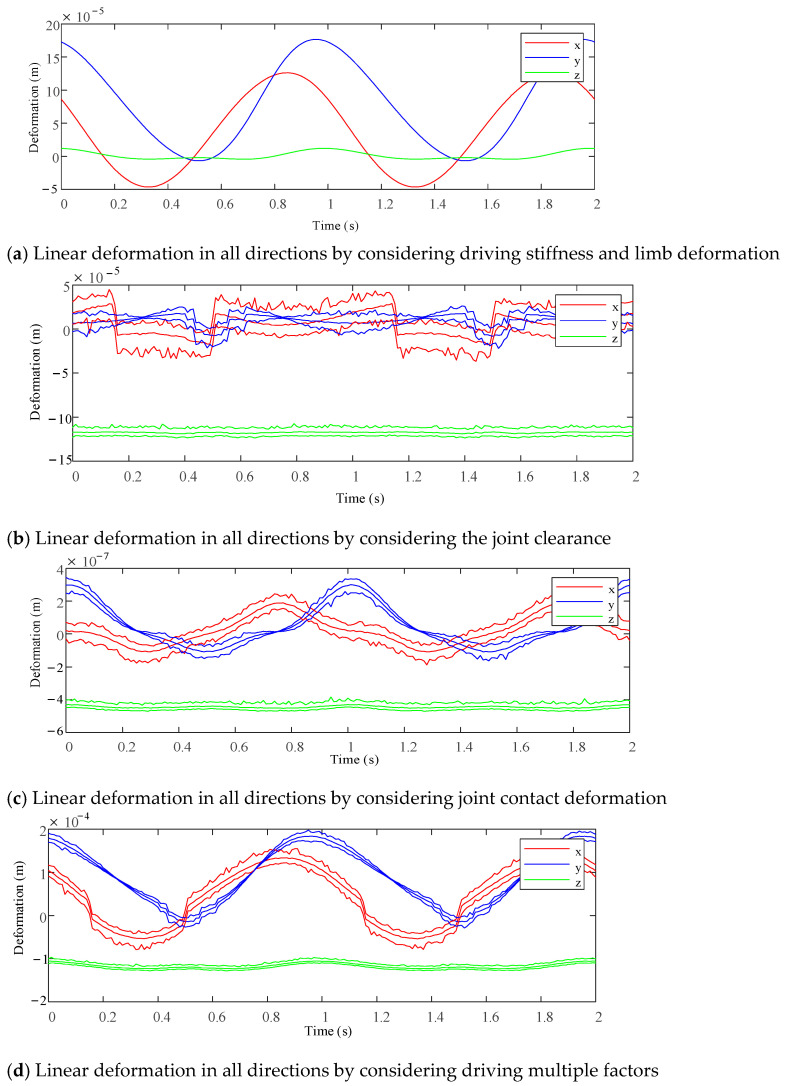
Linear deformation prediction interval of each direction at the end of the test platform under different factors when planning attitude and load.

**Figure 18 sensors-23-05916-f018:**
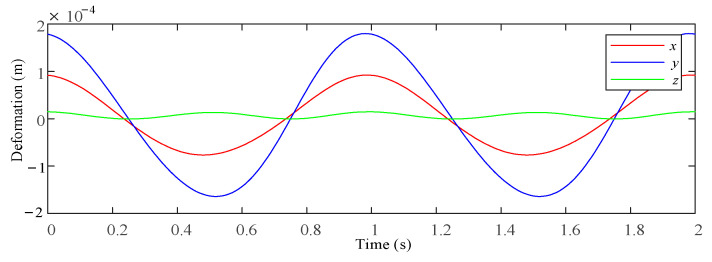
Linear deformation prediction interval of each direction at the end of the test platform under different factors when planning attitude and load 2.

**Table 1 sensors-23-05916-t001:** Theoretical and simulation comparison of 2UPR-RR-2RPU redundant PM with ideal joint.

Posture	Method	$\Delta X_{\mathrm{dx}}^{\varsigma }$/mm	$\Delta X_{\mathrm{dy}}^{\varsigma }$/mm	$\Delta X_{\mathrm{dz}}^{\varsigma }$/mm	$\Delta X_{\varphi x}^{\varsigma }$/°	$\Delta X_{\varphi y}^{\varsigma }$/°	$\Delta X_{\varphi z}^{\varsigma }$/°
1	Theory value	0.0232	0.1460	−0.00036	0.00 72	0.0011	0.0652
	Simulation value	0.0213	0.1387	−0.00039	0.0075	0.0012	0.0932
	error/%	8.92	5.26	7.69	4.00	8.33	8.58
2	Theory value	−0.0920	0.1176	0.0184	0.0058	−0.0058	0.0729
	Simulation value	−0.0968	0.1126	0.0170	0.0055	−0.0061	0.0790
	error/%	4.96	4.44	8.23	5.45	4.92	7.72

**Table 2 sensors-23-05916-t002:** Comparison of stiffness modeling theory and simulation of 2UPR-RR-2RPU redundant PM.

	Load 1			Load 2		
	*x*	*y*	*z*	*x*	*y*	*z*
Maximum stiffness considering joint clearance/mm	0.0226	0.0139	−0.1120	0.1816	0.1819	−0.0990
Minimum stiffness considering joint clearance/mm	0.0012	0.0128	−0.1208	0.1237	0.1524	−0.0844
Range of interval/mm	0.0214	0.0011	0.0088	0.0579	0.0295	0.0146
Sample mean value/mm	0.0130	0.0134	−0.1170	0.1543	0.1689	−0.0920
Maximum stiffness considering joint contact/μm	0.0493	0.0200	−0.4207	0.0964	0.1442	−0.0483
Minimum stiffness considering joint contact/μm	−0.0298	0.0181	−0.4636	0.0847	0.1337	−0.0417
Range of interval/μm	0.0150	0.0194	−0.4462	0.0117	0.0105	0.0066
Sample mean value/μm	0.0150	0.0194	−0.4462	0.0908	0.1391	−0.0447
Finite element simulation/mm	0.0129	0.0128	−0.1106	0.1616	0.1706	−0.0927
Error/%	0.78	4.69	5.79	4.51	0.99	0.75

## Data Availability

Not applicable.

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
