# Peer review of "Research on High Precision Stiffness Modeling Method of Redundant Over-Constrained Parallel Mechanism"

_sensors, 2023, doi:10.3390/s23135916_

Round 1
Reviewer 1 Report
This study has good significance, but its English expression needs to be improved. In view of the research problems, a clear experimental scheme should be given, and comparative analysis should be carried out before the experimental results.
Author Response
Thank you for the reviewers’ comments concerning our manuscript entitled “Research on high precision stiffness modeling method of redundant parallel mechanism with over constraints”(Sensors-2268338). Those comments are valuable and very helpful. We have read through comments carefully and have made corrections.
The stiffness model proposed in this paper is based on certain assumptions, simplifying the branch chain push rod and swing rod as circular cross-section columns, and simplifying the clearance hinge as axis and outer cylinder. Considering the axial and radial clearances of all U and R pairs. Simulation can monitor the contact status of hinges in real-time. Therefore, the correctness of the stiffness modeling method proposed in this paper can be verified by comparing simulation results with theoretical results.
At present, there is no design for the parallel mechanism prototype. The next step scheme will be to develop an experimental prototype and conduct stiffness performance experiments. The experimental scheme will be provided as follows:
(1) An experimental prototype is designed and a control system is built. The loads shown in Equations (97) and (98) in the paper are applied at the end of the moving platform. The displacement sensor and attitude sensor are installed at the center of the moving platform. During the assembly process, the axial clearance of all hinges of the parallel mechanism is initially 0.1mm, and the radial clearance is 0.05mm.
(2) Firstly, based on the closed-loop vector method, an error model of a parallel mechanism considering the driving stiffness, branch deformation, joint clearance, and joint contact deformation is established, and error analysis is conducted; Then, based on particle swarm optimization algorithm for parameter identification, the error model is embedded into the control system for calibration experiments to correct stiffness errors; Finally, the stiffness of the moving platform were measured using displacement sensor and attitude sensor, and compared with theoretical and simulation results for verification.
The English expression in the paper has been improved and marked in green.

Reviewer 2 Report
This paper proposed a high precision stiffness modeling method of redundant parallel mechanism with over-constraints. Based on the probability statistical model, the uncertainty of the offset value of the clearance joint and the contact area of the joint caused by the coupling of the branch constraint force is solved. The work is technically sound and the paper is well organized. However, there are several issues in this paper which should be revised to help improve the paper before accepting.
1. What are the deficiencies of the existing stiffness modeling method need to be supplemented in the abstract.
2. In the introduction of the paper, the review logic of this field is relatively chaotic, and it does not review the current situation from the perspective of historical logic or classification logic.
3. In section 5.1 of the paper, why 7500N external load is applied in the normal direction of the moving platform? Please give the application background and actual working conditions of the redundant parallel mechanism.
4. In line 265 of the article, the symbols of the kinematic pair of 2UPR-RR-2RPU redundant parallel mechanism should be roman type, and relevant errors need to be checked in full paper and corrected.
5. Please check the torque unit in formula (94).
6. In the conclusion, it is pointed out that the influencing extent of each factor on the end deformation is analyzed, and the specific factors of each factor need to be clearly described.
Author Response
Thank you for the reviewers’ comments concerning our manuscript entitled “Research on high precision stiffness modeling method of redundant parallel mechanism with over constraints”(Sensors-2268338). Those comments are valuable and very helpful. We have read through comments carefully and have made corrections.
- What are the deficiencies of the existing stiffness modeling method need to be supplemented in the abstract.
Author response: The above issues have been revised in the paper and marked in yellow. See lines 9-10 for the modified part.
- In the introduction of the paper, the review logic of this field is relatively chaotic, and it does not review the current situation from the perspective of historical logic or classification logic.
Author response: The above issues have been revised in the paper and marked in yellow. See lines 34-37、41-42、47-63 for the modified part.
- In section 5.1 of the paper, why 7500N external load is applied in the normal direction of the moving platform? Please give the application background and actual working conditions of the redundant parallel mechanism.
Author response: The above issues have been revised in the paper and marked in yellow. See lines 751-756 for the modified part.
- In line 265 of the article, the symbols of the kinematic pair of 2UPR-RR-2RPU redundant parallel mechanism should be roman type, and relevant errors need to be checked in full paper and corrected.
Author response: The above issues have been revised in the paper and marked in yellow. See lines 267 for the modified part.
- Please check the torque unit in formula (94).
Author response: The above issues have been revised in the paper and marked in yellow. See lines 793、828、830 for the modified part.
- In the conclusion, it is pointed out that the influencing extent of each factor on the end deformation is analyzed, and the specific factors of each factor need to be clearly described.
Author response: The above issues have been revised in the paper and marked in yellow. See lines 916 for the modified part.
The English expression in the paper has been improved and marked in green.
